# Infusing Theory of Mind into Socially Intelligent LLM Agents

## Abstract

Theory of Mind (ToM)—an understanding of the mental states of others—is a key aspect of human social intelligence, yet, chatbots and LLM-based social agents do not typically integrate it. In this work, we demonstrate that LLMs that explicitly use ToM get better at dialogue, achieving goals more effectively. After showing that simply prompting models to generate mental states between dialogue turns already provides significant benefit, we further introduce ToMAgent (ToMA), a ToM-focused dialogue agent. ToMA is trained by pairing ToM with dialogue lookahead to produce mental states that are maximally useful for achieving dialogue goals. Experiments on the Sotopia interactive social evaluation benchmark demonstrate the effectiveness of our method over a range of baselines. Comprehensive analysis shows that ToMA exhibits more strategic, goal-oriented reasoning behaviors, which enable long-horizon adaptation, while maintaining better relationships with their partners. Our results suggest a step forward in integrating ToM for building socially intelligent LLM agents.[1]

## 1 Introduction

Success in social interactions – defined by goal achievement, adherence to social norms, and more – depends not just on expressing our own intentions and beliefs, but also on understanding our conversation partners. Theory of Mind (ToM), the cognitive ability to understand the mental states of others (Premack & Woodruff, 1978; Baron-Cohen et al., 1985), captures this intuition and allows social reasoning and strategic behavior (Apperly & Butterfill, 2009). Here, we study whether ToM can serve as a similarly powerful element in social LLM agents.

The extent to which LLMs already possess ToM is debatable (Kosinski, 2024; Shapira et al., 2024), despite the deployment of LLMs in settings where understanding the user is crucial (e.g. job interviews, customer service). Methods for improving LLMs' ToM abilities range from chain-of-thought prompting (Wilf et al., 2024; Shinoda et al., 2025), through neuro-symbolic methods that combine LLMs with symbolic belief tracking (Sclar et al., 2023), to Bayesian Inverse Planning (Ying et al., 2023), and inference-time hypothesis generation (Kim et al., 2025). However, past work on ToM for LLMs typically evaluates this ability directly on QA setups (Kim et al., 2023; Chen et al., 2024), rather than its usefulness in social situations. Meanwhile, existing research in interactive social environments like Sotopia (Zhou et al., 2024) has largely focused on training models to generate utterances that lead to successful conversations (Kong et al., 2025; Yu et al., 2025), overlooking the role of explicit mental state modeling.

In this work, we address the question of *how to equip LLMs with Theory of Mind abilities that can effectively improve their social reasoning.* We demonstrate that even simply prompting LLMs to generate mental states between dialogue turns can significantly contribute to goal achievement. To maximize this benefit, we propose ToMAgent (ToMA), a method for goal-oriented social reasoning in dialogues that combines ToM predictions with conversation outcome prediction to select the best trajectory for training. As illustrated in Figure 1, given a social scenario such as "Two friends are camping in the cold and there is only one blanket" and opposing agent goals (e.g., Agent$_1$ wants to keep it for themselves while Agent$_2$ wants to share), the target agent (Agent$_1$) is asked to (i) make multiple hypotheses about the other agent's mental states, (ii) generate the corresponding next

---

[1] The code, training data, and models of this work will be publicly released.

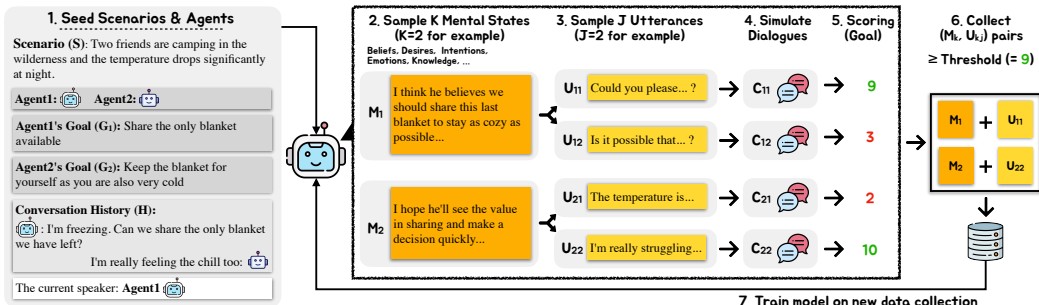

Figure 1: Overview of ToMA. We sample scenarios, goals, and conversation histories from Sotopia-Pi (Step 1), generate candidate mental state–utterance pairs and simulate dialogues (Steps 2–3), evaluate goal achievement to select high-utility pairs (Step 5), and train the model after collecting these training pairs (Steps 6–7).

utterances, and (iii) simulate the remaining dialogue and estimate the likelihood that each dialogue leads to goal completion. We then use the most successful conversations to fine-tune the same LLM to generate the partner's mental states (e.g., they are cold and uncomfortable) and the strategic utterances that are likely to result in goal achievement (e.g., suggesting a compromise).

ToMA is evaluated on the Sotopia dataset (Zhou et al., 2024; Wang et al., 2024), an open-ended social reasoning environment that includes diverse goal-oriented social scenarios such as collaboration, negotiation, persuasion, and competition. Our experimental results demonstrate that ToMA achieves score improvements by up to 18.9% and 6.9% compared to the best base model variant for Qwen2.5-3B and Qwen2.5-7B, respectively, and is also competitive with a GPT-5 nano baseline. Furthermore, we provide a comprehensive analysis of our results, including the success and failure factors across different scenarios and the ToM dimensions that are generated by the model. The analysis shows that ToMA exhibits more strategic, goal-oriented, and long-horizon behavior than the baselines, while also achieving better personal relationships with the partner. Our findings highlight that social reasoning in LLMs cannot be achieved through optimizing their performance on general reasoning benchmarks (Leaderboard, 2025) alone; it requires explicit modeling of mental states to enable safe, fair, and effective interactions with humans.

## 2 METHODOLOGY

In this section, we introduce ToMA, a look-ahead training framework that improves agents' ToM ability in social interactions to achieve their goals. Conditioned on a scenario (e.g., two friends are camping in the cold and there is only one blanket in Figure 1) and the agents' private goals (e.g., sharing the only blanket available vs. keeping the blanket for yourself), the goal is to reach a mutually agreeable solution, such as taking turns or sharing the blanket, through dialogue.

Our proposed training protocol consists of generating training examples and fine-tuning an LLM-based agent, as illustrated in Figure 1. First, we sample conversation contexts (§2.1). At each step of the dialogue, we use an LLM to first elicit multiple ToM hypotheses corresponding to the mental state of each agent (i.e., self and first-order beliefs), and then generate an appropriate utterance conditioned on these mental states (§2.2). To identify useful mental states and utterances that eventually contribute to goal achievement, we run short-horizon simulations and keep pairs that achieve the highest score on the simulated conversations (§2.2). Finally, we use the identified set of mental states and utterance pairs as training examples for fine-tuning the LLM to generate both the latent mental states and utterances (§2.3).

### 2.1 SAMPLING CONVERSATIONS TO SEED SCENARIOS AND AGENTS

To train models capable of socially grounded, goal-oriented reasoning in diverse contexts, it is imperative to use data that captures the complexity of real-world social interactions. To this end, we adopt the Sotopia-Pi dataset (Wang et al., 2024), which provides a diverse set of scenarios and social goals, allowing us to simulate complex social interactions during training. We first randomly sample 500 episodes from Sotopia-Pi, where each episode provides a social scenario, two agents with their

own goals, and a multi-turn dialogue between them. Then, for each scenario, we randomly sample two conversations provided by Sotopia-Pi and truncate each to at most four turns to ensure the context is early enough that the social goals have not yet been achieved. We denote each resulting instance, comprising a scenario, agents' social goals, and a partial conversation history, as $H$, which is referred to as the *context* in subsequent steps. These contexts serve as the default input set for eliciting useful mental states and utterances.

## 2.2 GENERATING AND SCORING ToM HYPOTHESES AND UTTERANCES

The goal of this phase is to generate plausible mental states and utterances that help an agent advance its own social goal, which can be used to train goal-oriented ToM-aware agents. Specifically, we ask the target model (Agent$_1$ in Figure 1), which is the model to be trained, to generate its own latent ToM states, produce corresponding utterances, and utilize these pairs for training.

**Exploring mental states and utterances.** The first key steps (2–3 in Figure 1) toward socially intelligent behavior is to explore a range of plausible mental states and corresponding utterances that align with the agent's social goals and conversational context. For this purpose, from each context $H$, which includes the scenario, the agents' private social goals, and the partial conversation history up to that point, we prompt an $LM_{target}$ to generate $K$ mental state hypotheses, where each hypothesis may consist of multiple sentences capturing different aspects of the current (target) agent's internal state: $m_k \sim LM_{target}(m \mid H)$. The model is asked to ensure that each generated hypothesis covers at least three out of the five ToM dimensions: *beliefs, desires, intentions, emotions,* and *knowledge*. For each mental state hypothesis $m_k$, we sample $J$ utterances: $u_{k,j} \sim LM_{target}(u \mid m_k, H)$. This gives us a candidate set of mental state and utterance pairs $\mathcal{C}_H = \{(m_k, u_{k,j})\}_{k=1..K,\ j=1..J}$.

**Running simulations to evaluate downstream utility.** To identify the most useful mental state and utterance pairs for training that most effectively contribute to successful goal achievement, we perform a short-horizon simulation to look ahead into the future trajectory of the dialogue and assess how each pair influences the goal achievement of agents throughout the conversation (Steps 4–5 in Figure 1). In the first turn, the target agent produces utterance $u_{k,j}$ conditioned on the mental state hypothesis $m_k$ and the context $H$. Then the conversation continues for up to four future turns, simulating the partner agent using $LM_{partner}$. Once the simulation is done, we compute the goal achievement score (0–10) for each agent, $S_{target}$ and $S_{partner}$, reflecting the degree to which each agent successfully advanced its objectives. Since a successful conversation is supposed to contribute to both agents' goals, the average goal score is calculated: $\hat{S}(h, m_k, u_{k,j}) = \frac{1}{2}(S_{target} + S_{partner})$. We retain all pairs with an average score $\geq 9$. If none meet this threshold, we keep the top-scoring pair. The resulting high-scoring pairs form a training set that we use for fine-tuning. See Appendix C for the prompts and the training instance format.

## 2.3 FINE-TUNING ON ToM STATES AND UTTERANCES

To instill Theory of Mind reasoning into the model, we fine-tune it on high-scoring mental state and utterance pairs identified through dialogue simulation that are maximally useful to advance their goals (Step 7 in Figure 1). From each selected pair $(m^\star, u^\star)$ and its context $H$ (i.e., scenario, private goal, and dialogue history), we construct two types of training examples: one where the model is prompted with $H$ and trained to generate $m^\star$ (i.e., *mental-state prediction*), and another where the model is prompted with both $H$ and $m^\star$ to generate $u^\star$ (i.e., *utterance prediction*). Together, we train the model to align with the joint behavior $P(u, m \mid H) = P(u \mid m, H) \cdot P(m \mid H)$ that led to high goal scores. We finetune the model $LM_{target}$ using a standard cross-entropy loss over next-token prediction. The resulting objective can be formalized as:

$$\mathcal{L}_{CE}(\phi) = \mathbb{E}_{(H, m^\star, u^\star) \sim \mathcal{D}^\star}\left[ CE(m^\star, \phi(H)) + CE(u^\star, \phi(H, m^\star)) \right] \quad (1)$$

$$= -\log P_\phi(m^\star \mid H) - \log P_\phi(u^\star \mid H, m^\star), \quad (2)$$

where $CE(y, \phi(x))$ denotes the token-level cross-entropy loss for target $y$ given input $x$ under model $\phi$. This way, the model learns to associate contexts with latent mental states and utterances that were empirically effective during simulation. This implicitly improves its internal mechanism over $P(m \mid H)$ and $P(u \mid m, H)$, aligning them to achieve their goals in various social situations.

## 3 EXPERIMENTAL SETUP

We follow the setup defined in Sotopia (Zhou et al., 2024). Each instance in Sotopia provides the scenario for the current social interaction between two agents, as well as their names and social goals. Models evaluated on Sotopia take the role of one agent, and they are tasked with having a dialogue with the other agent that results in achieving their own social goals. We describe the evaluation setup (§3.1) and training settings (§3.2). See Appendix A for more experiment details and Appendix C for all LLM prompts.

### 3.1 EVALUATION

**Data.**  We adopt the Sotopia-Eval dataset (Zhou et al., 2024), which provides multiple social scenarios for the agents to simulate conversations dynamically. We use both the `all` and `hard` sets to evaluate models. The `all` set includes 90 scenarios combined with 5 agent pairs, resulting in a total of 450 testing instances. Each pair among the five shares the same scenario description and agent goals, but the agent names and profiles are different. The `hard` set consists of 14 scenarios that are challenging to GPT-4 (Achiam et al., 2023), yielding 70 testing instances.

**Metrics.**  We follow Sotopia-Eval (Zhou et al., 2024), a suite of multi-dimensional evaluation metrics, and use LLM-as-a-Judge (Gu et al., 2024; OpenAI, 2025) to assess an entire conversation. We focus on the following central criteria from the original setup: (1) *Goal*: the extent to which the agent achieved their goals (0–10); (2) *Relationship* (Rel): whether the interactions between the agents help preserve or enhance their personal relationships prior to the conversation (-5–5); and (3) *Knowledge* (Know): whether the agent gained new and important information through the interaction (0–10). The LLM judge provides both the rating score and its rationale on each dimension and for each agent. We use GPT-5-mini (OpenAI, 2025) as the evaluator.

**Partner Agent.**  We follow the original Sotopia evaluation setup which evaluates both agents on their goal achievement and social awareness, and reports the average scores of the two agents. In this "Self-Play" setup, both agents are instantiated as a model with the same complexity (e.g., base with base, ToMA with ToMA, etc.).

**Baselines.**  We consider two base settings as follows: (1) *Base*: Using the vanilla language model (without fine-tuning), as a lower bound for the LLMs' ability to hold a social dialogue; and (2) *Base+MS*: where we apply a two-step prompt to the base model. We first generate mental states based on the context and then generate an utterance conditioned on the context and mental states. This setup quantifies both the quality and the utility of the mental states generated by the base model.

### 3.2 TRAINING

**Data.**  We use the scenarios and the agents' names and social goals from Sotopia-Pi (Wang et al., 2024) to seed our conversations, as shown in Figure 1, Step 1. We instantiate each agent with an instance of the pre-trained LLM (which we will later fine-tune on the training set described here). Then, we generate a conversation between the two agents using the simulation protocol provided by Sotopia, which defines the action types and schedules the agents to speak iteratively, and modify it to introduce mental states as a latent variable. Before generating each utterance, we prompt the agent to generate or update their own mental states and their first-order beliefs about the mental states of the other agent. We set the number of mental state hypotheses to $K = 2$ and the number of utterance candidates per hypothesis to $J = 2$.

**Models.**  We experiment with Qwen2.5-3B, Qwen2.5-7B (Qwen, 2024a), and LLaMA3.1-8B (Dubey et al., 2024) as the backbone LLMs. We use a 4-bit quantized version of Qwen2.5-14B as $LM_{partner}$ to ensure the partner generates reasonable utterances in simulations independent of the model size being trained. Finally, `Gemini-Flash` (Comanici et al., 2025) is used to score the simulated conversations.

**Fine-tuning.**  Utilizing the paired utterances (Uttr) and mental states (MS) from the generated multi-turn conversations, we conduct supervised fine-tuning (Pareja et al., 2025) over low-rank adapters (Hu et al., 2022) of small language models (i.e., Qwen2.5-3B and Qwen2.5-7B) with the data obtained in §2.2. We consider the following three training objectives: (1) *FT+Uttr*: Fine-tuning

models only on utterance generation, ablating the mental states supervision to assess its contribution to the conversation success; (2) **FT+MS**: Fine-tuning models to generate mental states, ablating the utterance generation to assess its contribution to the conversation success; and (3) **FT+MS+Uttr** (TOMA): Fine-tuning models on both utterance generation and mental states alignment, as explained in §2.3. For the evaluation of **FT+MS** and TOMA, the model generates mental states first and then produces the utterances to respect the causal constraint between the two.

## 4 EXPERIMENTS

We compare the performance of TOMA to the baselines (§4.1). Then, we analyze the effect of different partner agents on goal achievement (§4.2), the performance across scenario types (§4.3), and the success and failure factors in goal achievement (§4.4). Finally, we present a statistical analysis of TOMA's performance across different evaluation dimensions (Appendix B.1).

### 4.1 DOES THEORY OF MIND HELP WITH SOCIAL REASONING?

| Method | Qwen2.5-3B | | | | Qwen2.5-7B | | | | Llama3.1-8B | | | |
|---|---|---|---|---|---|---|---|---|---|---|---|---|
| | Rel | Know | Goal | Avg. | Rel | Know | Goal | Avg. | Rel | Know | Goal | Avg. |
| Base | 0.97 | 3.29 | 5.25 | 3.17 | 2.07 | 4.54 | 7.26 | 4.62 | 0.27 | 5.09 | 6.11 | 3.82 |
| Base+MS | 1.54 | 3.48 | 5.93 | 3.65 | 2.47 | 4.45 | 7.30 | 4.74 | 1.20 | 5.37 | 6.67 | 4.41 |
| FT+Uttr | 1.92 | 4.01 | 6.60 | 4.18 | 2.42 | **4.78** | 7.43 | 4.88 | 1.28 | 5.18 | 6.88 | 4.45 |
| FT+MS | **2.37** | 3.81 | 6.69 | 4.29 | **2.73** | 4.40 | 7.46 | 4.86 | 1.49 | 4.70 | 6.46 | 4.22 |
| FT+MS+Uttr (TOMA) | 2.18 | **4.22** | **6.84** | **4.41** | 2.70 | 4.77 | **7.67** | **5.05** | **2.37** | **5.61** | **7.48** | **5.15** |

Table 1: Overall performance in terms of Rel, Know, and Goal dimensions on the `all` split.

| Method | Qwen2.5-3B | | | | Qwen2.5-7B | | | | Llama3.1-8B | | | |
|---|---|---|---|---|---|---|---|---|---|---|---|---|
| | Rel | Know | Goal | Avg. | Rel | Know | Goal | Avg. | Rel | Know | Goal | Avg. |
| Base | 0.18 | 4.20 | 4.96 | 3.11 | 0.58 | 4.21 | 5.26 | 3.35 | -1.59 | 5.10 | 4.22 | 2.58 |
| Base+MS | 1.04 | 4.05 | 5.27 | 3.45 | 2.17 | 4.51 | 5.86 | 4.18 | -0.52 | 5.16 | 4.80 | 3.15 |
| FT+Uttr | 1.22 | 4.10 | 5.23 | 3.52 | 1.36 | 4.43 | 5.70 | 3.83 | -0.35 | 4.91 | 4.85 | 3.13 |
| FT+MS | 1.70 | 4.08 | 5.42 | 3.73 | **2.40** | 4.33 | 6.30 | 4.34 | 0.33 | 5.04 | 5.06 | 3.48 |
| FT+MS+Uttr (TOMA) | **1.90** | **4.22** | **5.88** | **4.00** | 2.33 | **4.78** | **6.32** | **4.48** | **1.27** | **5.36** | **5.68** | **4.10** |

Table 2: Overall performance in terms of Rel, Know, and Goal dimensions on the `hard` split.

**TOMA outperforms the baselines.** Tables 1 and 2 present the performance of models on the `all` and `hard` subsets of the Sotopia test set, respectively. On both subsets, TOMA consistently outperforms all other model variants across the relationship, knowledge, and goal completion dimensions. Moreover, TOMA performs competitively with a strong GPT-5-nano baseline (Base+MS in Table 3), even though GPT-5-nano surpasses Qwen2.5-7B on several general reasoning bench-

| Method | Rel | Know | Goal | Avg. |
|---|---|---|---|---|
| Base | 0.77 | 4.39 | 6.24 | 3.80 |
| Base+MS | **1.51** | **5.21** | **6.67** | **4.46** |

Table 3: Performance of GPT5-nano in terms of Rel, Know, and Goal dimensions on the `hard` split.

marks (White et al., 2025; Leaderboard, 2025). Specifically, TOMA (and even slightly more FT+MS) substantially outperforms GPT-5 nano on the relationship dimension, indicating it generates utterances with better sensitivity to the other partner's feelings. Compared to the best base model variant (Base+MS), TOMA achieves score improvements of 16.8%, 6.6%, and 23.45% on both datasets for the Qwen2.5-3B, 7B, and Llama-3.1 models, respectively, averaged across the `all` and `hard` sets.

**Mental-state conditioning improves relationship modeling.** We observe that models that generate utterances without explicit mental-state conditioning (Base and FT+Uttr) perform significantly worse on the relationship dimension than models that use mental-state representations (Base+MS, FT+MS, and TOMA). This may suggest that explicitly considering the partner agent's mental state can help the target agent preserve a positive relationship with them. Training on utterances alone (FT+Uttr) generally improves the knowledge and goal scores compared to the Base models on `all` split. The improvement in goal completion is unsurprising given that our fine-tuned models are

supervised to maximize goal completion. However, this goal-directed behavior may come at the expense of interpersonal sensitivity, as indicated in its lower relationship scores compared to models conditioned on mental states.

**Fine-tuning on mental states does not hurt utterance effectiveness.** Training the model only on mental states (FT+MS) could potentially decrease its general generation ability. However, our fine-tuned model is still able to produce reasonably effective utterances, achieving higher goal and relationship scores than the base models across both splits. ToMA, trained to jointly improve the prediction of latent mental states and the corresponding appropriate utterances, achieves the best of both worlds, effectively maintaining relationships, knowledge seeking, and goal-oriented behavior.

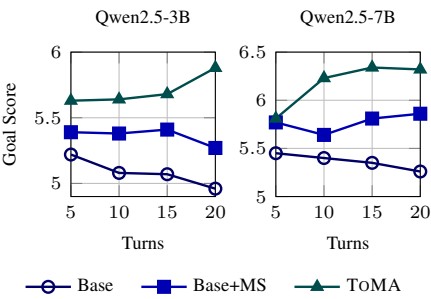

Figure 2: Goal completion scores across 5–20 turns on the `hard` split.

**Theory of Mind enables long-horizon adaptation.** Figure 2 compares how efficiently agents achieve their goals under different maximum turn limits. Surprisingly, the goal score of Base decreases as the number of turns increases. This is likely because the base model often repeats the same argument, making no progress across turns, which the GPT-5 judge penalizes. Base+MS shows slight improvement, but starts declining again for conversations longer than 15 turns. In contrast, ToMA consistently improves its goal completion score as the number of turns increases, suggesting that it may be adapting its strategy over time to achieve the goal more effectively. This adaptivity and long-horizon planning behavior can make ToM-informed agents better suited for real-world social interactions that often require longer and more flexible responses.

## 4.2 How do Different Partners Affect Goal Achievement?

Our main evaluations follow the original "self-play" setup where both agents are instances of the same model (e.g., ToMA with Qwen2.5-3B). Here, we address the question of how a different partner can impact the performance of the target agent. To that end, we test how a target agent based on the best model variants of each of base (Base+MS) and ToMA fares when paired

|  | Metric | Target Model | (Target=3B) Partner=Base- | | | | (Target=7B) Partner=Base- | | | |
|---|---|---|---|---|---|---|---|---|---|---|
|  |  |  | 3B | 7B | 14B | 32B | 3B | 7B | 14B | 32B |
| Both | Goal | Base+MS | 4.81 | **4.99** | 5.11 | 5.28 | 4.94 | 5.27 | 5.72 | 5.83 |
|  |  | ToMA | **5.00** | 4.96 | **5.36** | **5.40** | **5.23** | **5.41** | **5.75** | **5.86** |
|  | All | Base+MS | 3.17 | 3.29 | 3.4 | 3.49 | 3.38 | 3.56 | 3.79 | 3.93 |
|  |  | ToMA | **3.35** | **3.41** | **3.64** | **3.73** | **3.53** | **3.67** | **3.86** | **4.01** |
| Target | Goal | Base+MS | 3.85 | **4.35** | 3.58 | **3.63** | 3.84 | **4.35** | 4.12 | 3.77 |
|  |  | ToMA | **4.01** | 3.95 | **3.64** | 3.48 | **4.39** | 4.34 | **4.27** | **4.14** |
|  | All | Base+MS | 2.76 | **3.1** | 2.88 | 2.92 | 2.93 | 3.2 | 3.26 | 3.18 |
|  |  | ToMA | **2.96** | 3.04 | **3.02** | **3.1** | **3.18** | **3.25** | **3.28** | **3.42** |

Table 4: Performance of the target agent (Target) and average performance of both agents (Both) with respect to goal completion (Goal) and the average across goal, relationship, and knowledge scores (All). We use the `hard` split and vary the size of the partner agent (Base).

with a partner model of different complexity (Base) and size (3–32B). We conduct the evaluation on the `hard` split. For each scenario, we use the original 5 distinct role pairs and swap the agent roles (e.g., agent 1 as target and agent 2 as partner, and vice versa), resulting in 10 role pairs. We report both the goal completion score as well as the average across goal, relationship, and knowledge scores; once for the target agent and once for the average of both agents, in Table 4.

**A ToMA target agent not only improves its own goal completion, but also their partner's.** The target agent trained with our method performs best across most settings (Table 4). ToMA results in consistently better combined outcomes (Table 4, top) between target and partner, suggesting that our agent with improved ToM ability not only benefits itself, but also helps the other agent, likely reaching agreeable solutions for both agents. As we show in §4.4, this effect is likely due to the agent's ability to employ more effective strategies across a broader range of interaction scenarios (e.g., coordination, negotiation, persuasion, etc.). The individual outcome for the target is somewhat more complex (Table 4, bottom). For the larger target size (7B), ToMA results in consistently better target outcomes. For the 3B target size, the winner on goal achievement is inconsistent between ToMA and Base+MS; we hypothesize that it's harder for a small target agent to achieve their goal

when conversing with a larger and socially unaware partner. With that said, it is worth noting that ToMA wins at "All" metrics in most cases, meaning it is less likely than Base+MS to sacrifice relationships or knowledge.

**Coordination dynamics depend on both agent and partner sizes.** We observe that when the partner is larger, the overall conversation outcome – as measured by the average scores for both agents – improves. Looking at the target agent scores shows that the factors behind this improvement differ between the 3B and 7B ToMA target agents. The 7B target agent shows consistent performance improvement with partner size across all dimensions, suggesting that it can benefit from a more powerful partner. Conversely, the scores for the 3B target agent don't consistently improve with the partner's size, again suggesting that in that case, a larger partner leads to higher scores primarily *for the partner*. We observe that the 3B ToMA agent is more likely to achieve its goal when paired with an equal-size partner than with a considerably larger partner (14B or 32B).

### 4.3 How does ToMA Perform Across Different Conversation Types?

**Categorizing scenarios into types.** We are also interested in the performance and behavior of ToMA across different types of social interaction, where the agents' goals may be either aligned or competing. We manually examined the 90 scenarios in the `all` split and categorized them into four conversation types: **cooperation** - *a win-win situation where both agents can fully achieve their goals without conflicts or compromises* (36 scenarios); **negotiation** - *a positive-sum game where the agents can reach their goals to a satisfactory extent with certain compromises* (28 scenarios); **persuasion** - *a positive-sum game where the target agent tries to convince the partner to act in a way that promotes the target agent's goals* (13 scenarios); and **conflict** - *a zero-sum or even negative-sum game where their goals are in conflict and can hardly be solved through compromise* (13 scenarios). See Appendix B for full details of each scenario group.

**ToMA outperforms the base model under all scenario types.** We analyzed 450 conversations: five conversations for each of the 90 scenarios in the `all` split. Figure 3 looks at the average goal achievement score of the target agent in each conversation type, comparing agents implemented as the base model vs. ToMA. Data points on the right of the orange dotted line ($x = 0$ neutral line) correspond to conversations on which ToMA outperformed the base model. As observed, the first quartile (Q1) of each box is on the neutral line, indicating that ToMA outperforms base in at least 75% of the conversations of each type. Considering the inter-quartile range (IQR), ToMA brings greater gains in conflicts, where ToM may be more necessary for the target agent to achieve a goal that goes

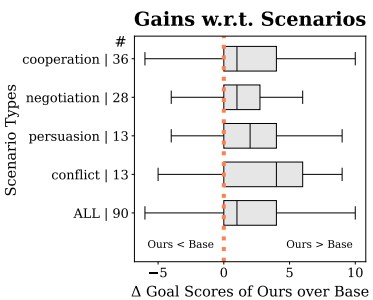

Figure 3: Performance gains of ToMA over Base w.r.t. scenarios.

against their partner. Furthermore, the lower boundary (Q1-1.5IQR) is about -5 while the upper boundary (Q3+1.5IQR) is nearly 10 (i.e., ToMA obtains an average score of 10 while the base model scores 0), showing that our method can largely outperform base, but not the other way around.

### 4.4 What Strategies Does ToMA Employ?

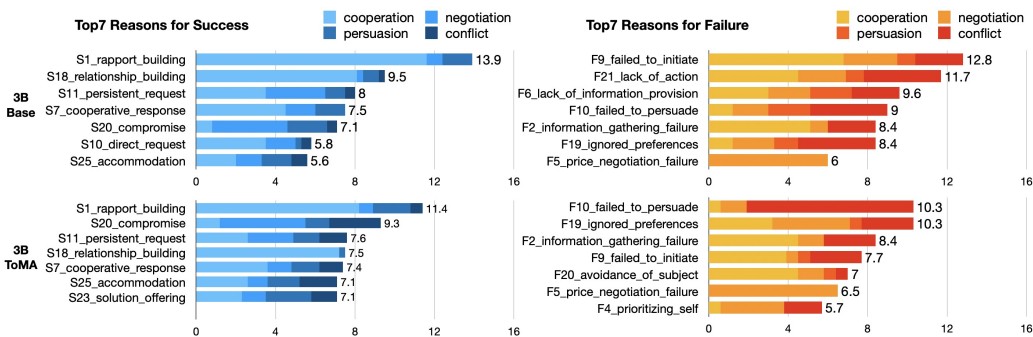

Figure 4: Top 7 goal success and failure factors for the Base model and, using the 3B model.

To understand the different strategies that agents with varying levels of ToM capabilities employ in order to achieve their goals, we analyze the factors contributing to successful conversations (goal completion score $\geq 7$) and the barriers leading to failed conversations (goal completion score $< 4$) across different model variants.

**Categorizing success and failure reasons.** To identify successful strategies, we provide Gemini with the full conversation, as well as the target agent's name and social goal, and prompt it to explain the reasons for success. Using the reasons from all the successful conversations, we prompt the LLM to categorize the reasons and provide a concise definition for each reason. To reduce redundancy, we further instruct the LLM to cluster and merge similar reasons into 25 representative ones, each manually verified by the authors for validity. Finally, we prompt the LLM to classify the reasons provided for each conversation into these canonical categories. We repeat the same process to obtain the failure reasons from the failed conversations.

Figure 4 presents the top factors most frequently associated with success and failure outcomes of the 3B models, with the respective prefixes `S_` or `F_`. Each label is further broken down by scenario types (Details in §4.3). See Appendix B for complete definitions of the labels and scenario categories.

**TOMA enables more strategic reasoning across diverse scenarios.** In successful conversations, the base model relies heavily on interpersonal strategies, such as `rapport building` and `relationship building`, and direct goal-pursuit approaches, such as `persistent request` and `direct request`. In contrast, TOMA adopts long-horizon goal-oriented strategic behavior by employing `compromise`, `accommodation`, and `solution offering`, while still maintaining comparable levels of `rapport building` and `cooperative response` to the base model.

In terms of conversation types, both models achieve success mainly in cooperative conversations, where it's easy for both agents to achieve a high goal completion score. Compared to the base model, TOMA also has high levels of success in competitive settings (negotiation, persuasion, and conflict), especially when using the strategies of compromise, accommodation, and solution offering. The results of the 7B model (in Appendix B.3) similarly show that TOMA applies strategic behaviors which lead to success across different scenario types, and this strategic behavior seems to increase with model size.

**TOMA exhibit more active behaviors in failure modes.** The base model often fails due to being too passive (`failed to initiate`; `lack of action`; `lack of information provision`). Conversely, TOMA employs active strategies that sometimes fail (e.g., `failure to persuade`) as well as goal-oriented approaches that fail to account for the role of relationship building in goal achievement (`ignored preferences`; `prioritizing self`). In the 7B version of TOMA, these failures are significantly reduced while the `lack of action` frequency is increased. We hypothesize this is the result of increased sensitivity to the partner's emotional state compared to the 3B model (as shown in the relationship score in Tables 1 and 2), which reduces the selfish `ignored preferences` and `prioritizing self` occurrences (see Appendix B.3).

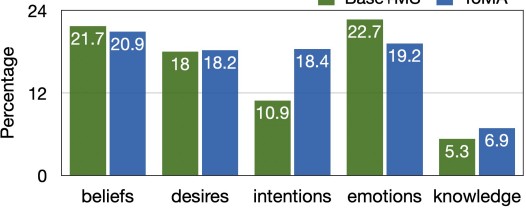

| Size | Model | 0th-order (%) | 1st-order (%) |
|---|---|---|---|
| 3B | Base+MS | **28.1** | 71.9 |
| | TOMA | 21.8 | **78.2** |
| 7B | Base+MS | **22.3** | 77.7 |
| | TOMA | 17.6 | **82.4** |

Figure 5: Distribution of mental state dimensions on the 3B model. See Appendix B.3 for 7B.

Table 5: Zero- vs. first-order reasoning percentage on Base+MS and TOMA.

**TOMA prioritizes intentions over emotions in mental state generation.** To investigate the effect of TOMA across different mental states, we categorize the generated ToM hypotheses into five dimensions and then compare the mental states distributions given by Base+MS and TOMA. Figure 5 shows that TOMA generates more hypotheses about intentions and relies less on emotions, while maintaining similar levels for beliefs, desires, and knowledge. This is in line with the finding that the base model is focused on rapport-building strategies, which require hypothesizing about the other agent's emotions – as opposed to TOMA's strategic and goal-oriented behavior that requires

reasoning about the other agent's intentions. We observe similar trends in the 7B model. In addition, we present mental state distributions under different scenario types in Figure 12 and qualitative examples of the mental states in the conversation (see Appendix B.5), which provides further insights into how different mental state dimensions contribute to the success of TOMA.

**TOMA generates more 1st-order mental states than the baseline.** Table 5 shows the distribution of 0th-order (target agent's own beliefs) and 1st-order (target agent's beliefs about others) mental states generated by Base+MS and TOMA. Although both models are prompted to produce these states in equal proportions, TOMA consistently generates more 1st-order beliefs by an average of +6.3% and +5.0% on the 3B model and 7B model, respectively, compared to Base+MS. This suggests that TOMA is better at inferring others' mental states, potentially contributing to more strategic and socially aware behaviors during interaction.

## 5 FURTHER ANALYSIS

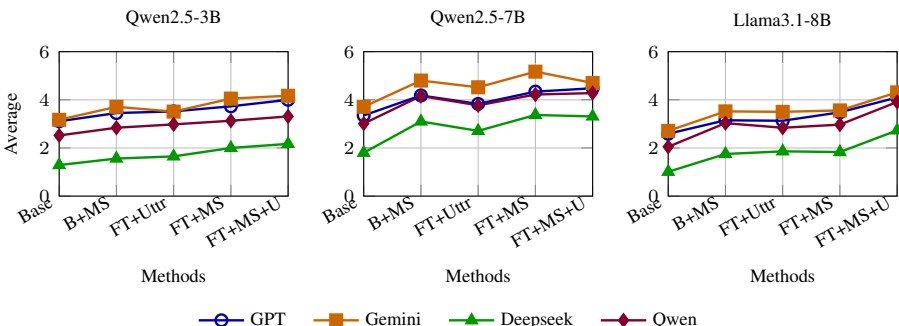

Figure 6: Average scores (relationship, knowledge, and goal) across 4 different LLM judges on the `hard` split. The trends of evaluation results remain consistent across different LLM judges. See Appendix B.4 for full results.

**Validity of the LLM-as-a-judge evaluation protocol.** To test whether our evaluation method is sensitive to the evaluator LLM, we experiment with three additional LLM judges: Gemini-2.5-flash, DeepSeek-3.1 (Liu et al., 2024), and Qwen3-225B, and report their scores along with those from the original evaluator GPT-5-mini. Figure 6 shows consistent trends across all four judges. TOMA consistently outperforms all baselines, while the SFT+MS model performs comparably when trained with the Qwen2.5-7B. Table 14 in the appendix provides detailed results, showing that TOMA improves the relationship dimension scores by an average of 1.58, 1.77, and 2.94 points over the Base on Qwen2.5-3B, Qwen2.5-7B, and LLaMA3.1-8B, respectively. For the goal dimension, TOMA achieves average gains of 1.06, 1.27, and 1.53 points over the Base model on the same three models. Furthermore, the ratings from the different LLM judges across all experimental settings are positively correlated (see Table 15 in the appendix), and our human validation shows that human evaluators validate the reasoning provided by the original GPT-5-mini judge (see Table 16 in the appendix).

**Performance across different K/J numbers and simulation turns.** Table 6 presents ablation experiments with different values of $K$ and $J$, as well as varying numbers of dialogue simulation turns. When varying $K$, we fix $J = 2$, and when varying $J$, we fix $K = 2$. For simulation turns, we vary the number of turns up to 8. Although all settings outperform the baselines, we observe some differences in their trends. Overall, increasing the number of simulation turns improves performance, but using 4 turns provides a good balance between data con-

| #MS | Avg. | #Uttr | Avg. | #Turn | Avg. |
|-----|------|-------|------|-------|------|
| 2 | 4.01 | 2 | 4.01 | 4 | 4.01 |
| 3 | 4.17 | 3 | 3.81 | 6 | 4.24 |
| 4 | 4.11 | 4 | 4.18 | 8 | 4.06 |

Table 6: Average scores across different numbers of mental states, utterances, and simulation turns on the `hard` split with Qwen2.5-3B model.

struction efficiency and overall effectiveness. Similarly, sampling more mental states and utterances (e.g., 3 or 4) yields modest improvements, although $J = 2$ remains sufficient; we encourage future work to explore more diverse settings.

## 6 RELATED WORK

**Theory of Mind in LLMs.** With the advent of LLMs, research on ToM in AI is experiencing strong momentum. Studying the extent that LLMs have ToM abilities can inform research on building AI agents with human-like communication and empathy skills, as well as protecting against AI manipulation and deception. Current findings are conflicting: LLMs achieve good performance on various ToM benchmarks and tests designed for humans, which some researchers interpret as having developed a theory of mind (Kosinski, 2023; 2024; Strachan et al., 2024); Yet others show that this ability is inconsistent and superficial (Ullman, 2023; Shapira et al., 2024; Amirizaniani et al., 2024; Nickel et al., 2024; Soubki & Rambow, 2025). To improve LLMs' ToM capabilities, one approach is to prompt models in a chain-of-thought setup to explicitly reason about beliefs and mental states before making a prediction (Wilf et al., 2024; Shinoda et al., 2025). Alternative approaches combine LLMs with belief tracking (Sclar et al., 2023; Qiu et al., 2024) or Bayesian Inverse Planning (Ying et al., 2023). While less brittle than pure LLM-based approaches, these methods are typically limited in scope and only applied to specific setups. Another promising (but computationally expensive) approach generates and explores multiple hypotheses about the agents' mental states during inference (Kim et al., 2025). In contrast, we propose a training approach that saves inference-time costs. Crucially, most existing work evaluates LLMs on static and artificial ToM benchmarks, requiring models to answer questions as an observer rather than a participant in a dynamic environment (Wagner et al., 2025; Xiao et al., 2025; Lupu et al., 2025). Instead, we evaluate our method on Sotopia, measuring the contribution of modeling ToM for social conversations between AI agents.

**Look-Ahead Simulation in Self-Training Agents.** In this work we leverage look-ahead, a planning technique where an agent simulates the potential outcomes several steps into the future to make more informed decisions in the present. In text generation, look-ahead search was employed for decoding, prioritizing tokens that lead to better overall generated text (Lu et al., 2022; Fu et al., 2024) or faster inference (Leviathan et al., 2023; Chen et al., 2023). More recently, look-ahead signals were used in GRPO (Guo et al., 2025), an RL algorithm used in LLM preference tuning. GRPO obviates the need for human-labeled data by generating multiple outputs, simulating their outcomes with an LLM-as-a-judge (Gu et al., 2024), and rewarding outputs that yield better outcomes. In general, many simulation-based methods focus on outcome alignment using RL (Xi et al., 2024; Pang et al., 2024). Conversely, we use simulation to generate training examples, similarly to Hoang et al. (2025). In the context of social dialogues, prior work targeting Sotopia employed a similar approach of generating conversations, simulating their outcome with an LLM judge (e.g., in terms of goal achievement), and using this signal to select positive training examples or as a reward in RL (Wang et al., 2024; Kong et al., 2025; Yu et al., 2025). Instead of directly optimizing utterances that lead to goal achievement or other desirable outcomes – which could lead to reward hacking – we explicitly train our model to use ToM in social dialogues; we improve both the model's ability to reason about mental states, as well as the capacity to consider this information when generating utterances.

## 7 CONCLUSION

We introduced TOMA, a training framework that integrates ToM-driven mental state and utterance prediction with conversation simulation to select interaction trajectories that best support goal achievement. Experiments on the Sotopia interactive evaluation benchmark demonstrate the effectiveness of our approach across a range of baselines, achieving competitive performance with GPT-5-nano. Comprehensive analysis demonstrates that TOMA, infused with ToM ability, can better infer others' mental states, leading to more strategic and goal-oriented behavior, as well as supporting long-horizon adaptation and improving relationship management. In conclusion, TOMA represents a significant step toward building socially intelligent LLM agents through explicit modeling of social reasoning and internal agent mechanisms.

## ETHICS STATEMENT

**Ethical considerations of social intelligence in LLMs.** LLMs are increasingly used as social partners, providing mental-health support, personalized guidance, and assistance in everyday decision-making. As these systems become embedded in human-AI interactions, understanding

their social behaviors becomes essential. Prior work showed that human-like social intelligence, such as empathy, can improve user experience and conversational quality (Campbell & Babrow, 2004; Shen, 2011; Chockkalingam et al., 2025). Our findings complement this line of work by demonstrating that explicitly modeling an interlocutor's mental state and conditioning the generation of utterances on these predictions improves both agents' relationship outcomes and goal achievement across diverse social scenarios. While such capabilities have clear benefits for supportive applications like tutoring, counseling, or customer service, they also introduce risks if exploited for manipulation or deception, such as in social media bots, political persuasion, or scams. Mitigating these risks requires public education about AI capabilities and risks, thoughtful regulation, and responsible design. In particular, we recommend that LLM-powered applications avoid human names or avatars and clearly identify themselves as AI systems to reduce the likelihood of misleading users.

**Data collection and ethics approval.** All procedures involving human participation were reviewed and approved by our institution's Research Ethics Board and adhered to all applicable institutional and federal guidelines. Human evaluations were conducted through CloudResearch, and all annotators provided informed consent. No personal information was collected at any stage, and participants were compensated at an average hourly rate of $10, which is comparable to the U.S. minimum wage.

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

# A EXPERIMENT DETAILS

## A.1 MODEL SETTINGS

For open-weight models such as Qwen (Qwen, 2024b), we load the model checkpoint and tokenizer provided by Hugging Face Transformers (Wolf et al., 2020). We load all models in the brain floating-point format (`bfloat16`). The maximum context length is set to 4096, random seed to 42, generation temperature to 0.7, and we use top-p sampling (Holtzman et al., 2020) with $p = 0.9$. For proprietary LLMs (GPT-5 (OpenAI, 2025) and Gemini (Comanici et al., 2025)), we call the respective API using a default generation temperature of 1.0. Table 7 provides the model sources.

| Type | Role | Model | Link |
|---|---|---|---|
| Open-weight LLM | Speaker (fine-tuning) | Qwen2.5-3B | Model Link |
| Open-weight LLM | Speaker (fine-tuning) | Qwen2.5-7B | Model Link |
| Open-weight LLM | Partner (frozen) | Qwen2.5-14B | Model Link |
| Proprietary LLM | Partner (frozen) | GPT-5-nano | API Link |
| Proprietary LLM | Evaluator (frozen) | GPT-5-mini | API Link |
| Proprietary LLM | Evaluator (frozen) | Gemini-Flash | API Link |

Table 7: The sources of models used in this work.

## A.2 TRAINING DETAILS

We adopt LoRA (Hu et al., 2022) for fine-tuning and apply grid search, provided by wandb (Biewald, 2020), on the learning rate and LoRA configurations (rank and alpha), and select the best model checkpoint based on the performance on the validation set. During validation, the model is evaluated on 20 randomly sampled testing instances and is asked to generate 10 turns of conversation per instance. In addition, we employ an early stopping strategy to end the training session when the best validation score does not change for 3 consecutive updates. The key training hyper-parameters are presented in Table 8.

| Hyper-parameters | Values |
|---|---|
| # epochs | 3 |
| batch size | 2 |
| gradient accumulation steps | 4 |
| learning rate | 1e-4; 5e-05 |
| lr scheduler | cosine |
| weight decay | 0 |
| warmup steps | 10 |
| max seq len | 4,096 |
| LoRA rank | 8; 16; 32; 64 |
| LoRA alpha | 32; 64; 128 |
| LoRA dropout | 0 |

Table 8: The training hyper-parameters.

## A.3 EXPERIMENTAL COSTS

For constructing the training data containing mental states and utterances, the API calls of Gemini (`gemini-2.0-flash-lite-001`) cost less than 5 USD. For the comprehensive evaluation in our experiments, the cost of GPT-5 (`gpt-5-mini`) was roughly 100 USD.

Each experiment session involving open-weight LLMs was conducted on a single NVIDIA L40S GPU, and we employ `unsloth` (Daniel Han & team, 2023) for fast training, reducing each training session to about 4 hours.

# B ANALYSIS DETAILS

## B.1 HOW DOES TOMA PERFORM ACROSS DIFFERENT EVALUATION DIMENSIONS?

To investigate the performance gains of TOMA over Base in different evaluation dimensions (i.e., Goal, Relationship, and Knowledge), we visualize the paired scores in Figure 7 and Figure 8, where each point (x, y) means the Base performance is x and TOMA performance is y for one instance. The 45-degree dot line ("neutral line") stands for a draw, and a darker color of the points represents a higher frequency.

We observe that more points are distributed above the neutral line, meaning TOMA outperforms Base for more instances, especially for Goal and Rel dimensions. In addition, considering the four quadrants of the Goal dimension in Figure 7(a) and Figure 8(a), many points lie in the upper-left region, meaning TOMA is much better than Base, while hardly any points lie in the lower-right corner. For the Relationship dimension in Figure 7(b) and Figure 8(b), most points of TOMA and Base are above the y=0 line, meaning a the relationship between two agents is preserved and even enhanced through after the conversation. Figure 7(c) and Figure 8(c) show that both methods help agents gain new or important information through interaction, and TOMA often brings more knowledge gains.

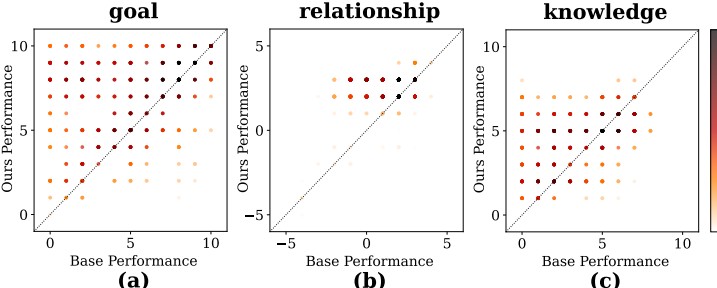

Figure 7: Comparisons between TOMA and Base over different dimensions using Qwen2.5-3B.

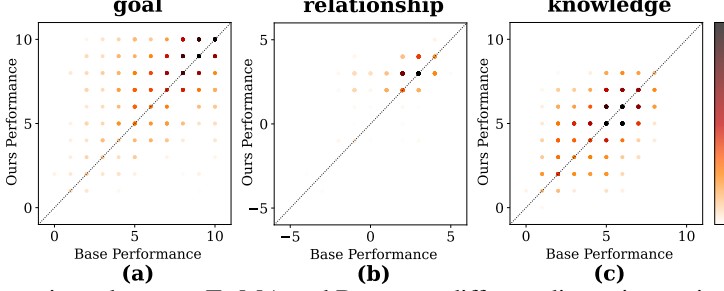

Figure 8: Comparisons between TOMA and Base over different dimensions using Qwen2.5-7B.

In addition, we present the correlation coefficients between the results of different dimensions in Table 9, which shows that the three dimensions are positively correlated with each other. We observe that the Goal-Rel pair shows the strongest correlation, indicating that the improved goal completion performance is related to the preservation and enhancement of the agents' relationship throughout the conversation, which supports the importance of enabling Theory of Mind.

|  | Goal–Rel | | Goal–Know | | Rel–Know | |
|---|---|---|---|---|---|---|
|  | **Pearson** | **Spearman** | **Pearson** | **Spearman** | **Pearson** | **Spearman** |
| 3B | 0.224 (2e-6) | 0.376 (5e-16) | 0.228 (1e-6) | 0.213 (7e-6) | 0.288 (9e-10) | 0.222 (3e-6) |
| 7B | 0.284 (2e-9) | 0.370 (2e-15) | 0.120 (0.013) | 0.136 (5e-3) | 0.107 (0.026) | 0.062 (0.195) |

Table 9: The Pearson and Spearman correlation coefficients (with p-values) between dimensions.

## B.2 HOW DOES TOMA PERFORM ACROSS DIFFERENT CONVERSATION TYPES?

Figure 9 provides the performance (Goal) gains of ToMA over Base with respect to different scenario types using the Qwen2.5-7B model, and the analysis of the 3B model (Figure 3) is described in §4.3. Table 10, Table 11, Table 12, and Table 13 showcase five instances per scenario type: cooperation, negotiation, persuasion, and conflict.

Figure 9: The Goal gains of ToMA over Base regarding different scenario types using Qwen2.5-7B.

| Scenario Description | Speaker's Goal | Partner's Goal |
|---|---|---|
| Conversation between two individuals in a romantic relationship | Ask the other to go to dinner with you (Extra information: you want to have a romantic dinner at a fancy restaurant) | Maintain a balanced diet while enjoying the dinner (Extra information: you are on a diet and you have a daily limit on calorie intake.) |
| Two friends are meeting at a coffee shop, where one of them is having trouble keeping up with their bills. | Help your friend with their financial trouble (Extra information: you are aware that your friend is struggling with their bills and you want to help.) | Maintain your pride if your friend offers you money(Extra information: you are the friend who is struggling financially, but you don't want to seem like you are taking advantage of the situation. Hint: you can express gratitude and discuss how you plan to pay it back in the future.) |
| Two friends are having a conversation in a coffee shop | Seek emotional support (Extra information: you have been feeling down lately and you need someone to talk to) | Provide emotional support but also avoid enabling victim mentality (Extra information: you believe that your friend needs to face their problems instead of dwelling on them.) |
| A conversation between two friends at a park | Break bad luck (Extra information: you found out that you have been having bad luck because you have been unknowingly carrying a cursed artifact.) | Help friend prepare the ritual to break the bad luck, although you are an atheist and not superstitious at all. (Extra information: you know that your friend believes in the ritual deeply, and the placebo effect might give them more confidence.) |
| Two strangers are meeting at a party. | You are trying to figure out whether you have a mutual friend with the other person. You should not simply list their names. | You are trying to figure out whether you have a mutual friend with the other person. You should not simply list their names. |

Table 10: Five instances of the *cooperation* scenario type.

| Scenario Description | Speaker's Goal | Partner's Goal |
|---|---|---|
| Two friends are camping in the wilderness and the temperature drops significantly at night. | Share the only blanket available (Extra information: you are extremely cold and need the blanket to keep warm Hint: you can stress the benefit of sharing.) | Keep the blanket for yourself as you are also very cold (Extra information: you have a lower tolerance for cold, and you feel that you need the blanket more) |
| One person is offering a BMW Z3 for $5000.0, while another individual is interested in purchasing it. Here's a brief description of the car: The BMW Z3 is in need of a good home. It operates smoothly without any problems and is known for its fuel efficiency and sporty appearance. The mileage is relatively low for its age. It's an opportunity not to be missed, so come and get it before it's gone. A smog certificate is readily available. | You are the buyer for this item and your target price is $4600. You should be aware that if you purchase it at a price significantly higher than your target, you will incur a penalty. However, if you manage to secure it for less than the target price, you will receive a bonus. | You are the seller of this item, with a target price set at $3260. Please note, you may face a penalty if this item is sold for a price significantly lower than the target. However, you stand a chance to earn a bonus if you successfully sell it for a price higher than the target. |
| Two roommates deciding on how to split up items after a garage sale. The items are 3 books, 2 hats, and 1 ball. Each item has a different sentimental value for each person, which translates into points. | Maximize the points you have (Extra information: you value the books at 3 points each, the hats at 2 points each, and the ball at 1 point) | Maximize the points you have (Extra information: you value the books at 2 points each, the hats at 3 points each, and the ball at 1 point) |
| Conversation between two friends, one who has written a play based on their parents' life, and the other owns a production company. | Sell the rights of the play to your friend (Extra information: you believe that the play is worth a lot and you would like a fair price for it) | Acquire the rights of the play while maintaining a budget (Extra information: you only have a limited budget for new plays this season. Hint: you can not spend all your budget on this play since you still need to maintain your financial stability.) |
| A conversation between two individuals, one is the homeowner and the other is their cousin's partner, who has been staying at the house for a while. | Tell the cousin's partner that they are no longer welcome in your home (Extra information: they have been causing a lot of trouble and not respecting house rules Hint: you may want to discuss the issues and propose possible solutions) | Attempt to remain in the home while maintaining a good relationship with the cousin and the homeowner (Extra information: you have nowhere else to go at the moment and you believe the issues can be resolved.) |

Table 11: Five instances of the *negotiation* scenario type.

| Scenario Description | Speaker's Goal | Partner's Goal |
|---|---|---|
| Two friends meet at a bar. | Convince the friend to drink less (Extra information: your friend has been drinking a lot recently and you're worried about their health) | You want to keep drinking but don't want to upset your friend (Extra information: you've been drinking to cope with stress recently, but don't want to worry your friend) |
| Two roommates living together and sharing household chores. One of them, who is responsible for cooking, finds out that the other one refuses to eat anything they cook | Convince the roommate to try the food (Extra information: you have spent a lot of time and effort on cooking) | Express your concerns about the food without hurting the roommate's feelings (Extra information: you are worried about the taste and nutrition of the food) |
| Two friends discussing their schedules at a coffee shop | Convince the friend to take more classes (Extra information: you think your friend is not challenging themselves enough) | Maintain a manageable schedule while preserving the friendship (Extra information: you are already overwhelmed with your current workload Hint: you can express your concerns about the workload and suggest other ways of challenging oneself.) |
| Conversation between two business partners reviewing their company's financial status (Gourmet Delights Inc. is globally recognized for producing and distributing a diverse line of exceptional quality food products.) | Convince the partner to cut costs (Extra information: you have found a report indicating the company's profits are decreasing. And you think the company should start by switching suppliers of ingredients. Hint: you can propose ways to cut costs that won't affect the status of the company.) | Maintain the quality of the company's products and services while agreeing to a cost cut (Extra information: you are concerned about the impact of cost-cutting on the quality of your product.) |
| A conversation between two individuals at a charity gala | Convince the other individual to donate to a cause supporting young children (Extra information: this cause can greatly improve the lives of many young children and you think it is a worthy cause. The minimum donation for this charity organization is $3000) | Maintain financial stability while supporting causes (Extra information: you have a certain budget for donations this year and have already made several donations. You are only thinking of donating $2000 max) |

Table 12: Five instances of the *persuasion* scenario type.

| Scenario Description | Speaker's Goal | Partner's Goal |
|---|---|---|
| Conversation between two friends at a trivia night | Get the answer for the trivia question from the friend (Extra information: your friend knows the answer to the trivia question but hasn't shared it with you yet) | Avoid revealing the answer to the trivia question (Extra information: You think that your friend should try to figure it out on their own to improve their trivia skills) |
| Two roommates are having a disagreement over noise levels in their shared apartment. | Convince your roommate to lower the noise (Extra information: you have an important online meeting tomorrow morning and you need a quiet environment to prepare) | Defend your right to play music in your own room, but also keep the peace in the apartment (Extra information: you find playing music relaxing and it's your way to unwind after a long day. Hint: you may want to suggest a compromise, like using headphones or playing music only during certain hours.) |
| Two friends are playing a video game together. | Win the game (Extra information: you just need five more minutes to accomplish your goal in the game) | Stop the game because it's too late (Extra information: you feel tired and need to go to bed early but don't want to upset your friend) |
| Conversation between two individuals who share a common dislike for a third person | Injure the third person (Extra information: You have a deep-seated hatred for the third person and you are willing to take the risk to hurt them) | Prevent physical harm but also express your dislike (Extra information: You also dislike the third person, but you are not willing to resort to violence, you prefer to express your dislike in a non-violent way. Hint: you can try to convince the other person about the potential consequences of engaging in violence) |
| Two criminals are arrested and placed in separate cells. The police have enough evidence to convict both criminals of a minor crime but need their confessions to establish evidence for a more serious crime. Now they get a chance to briefly chat with each during lunchtime. | Minimize the jail year (Extra information: your lawyer told you that 1. if you report the other person, you would be free, the other person would go to jail for 10 years, and vice versa. 2. if both of you remain silent, you will only receive a fine. 3. If both of you reported each other, you both would go to jail for 10 years. Hint: You may want to share your information to reach mutual understanding) | Minimize the jail year (Extra information: your lawyer told you that 1. if you report the other person, you would be free, the other person would go to jail for 10 years, and vice versa. 2. if both of you remain silent, you will only receive a fine. 3. If both of you reported each other, you both would go to jail for 10 years. Hint: You may want to share your information to reach mutual understanding) |

Table 13: Five instances of the *conflict* scenario type.

### B.3    WHAT STRATEGIES DOES TOMA EMPLOY?

Figure 10 presents the Top-7 goal success and failure reasoning labels on Base and TOMA on the Qwen2.5-7B model, and the reasoning of the 3B model (Figure 4) is described in §4.4. Table 17 and Table 18 provide the canonical labels for success and failure reasons, respectively. Figure 11 presents the distribution of mental state dimensions for 7B model.

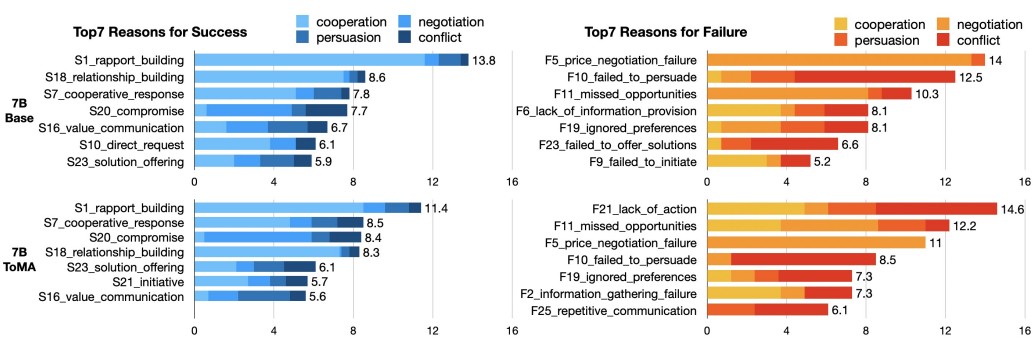

Figure 10: Top 7 goal success and failure reasoning labels on Base and TOMA on 7B model.

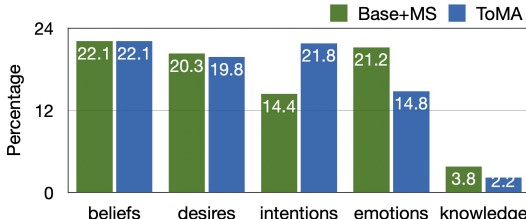

Figure 11: Distribution of mental state dimensions on 7B model.

### B.4    RESULTS OF MULTIPLE LLM JUDGES AND HUMAN EVALUATION

| | | Qwen2.5-3B | | | | Qwen2.5-7B | | | | LLaMA3-8B | | | |
|---|---|---|---|---|---|---|---|---|---|---|---|---|---|
| | | Rel | Know | Goal | Avg. | Rel | Know | Goal | Avg. | Rel | Know | Goal | Avg. |
| GPT5 | Base | 0.18 | 4.2 | 4.96 | 3.11 | 0.58 | 4.21 | 5.26 | 3.35 | -1.58 | 5.07 | 4.29 | 2.59 |
| | Base+MS | 1.04 | 4.05 | 5.27 | 3.45 | 2.17 | 4.51 | 5.86 | 4.18 | -0.52 | 5.16 | 4.8 | 3.15 |
| | SFT+Uttr | 1.22 | 4.1 | 5.23 | 3.52 | 1.36 | 4.43 | 5.7 | 3.83 | -0.35 | 4.91 | 4.85 | 3.13 |
| | SFT+MS | 1.7 | 4.08 | 5.42 | 3.73 | **2.4** | 4.33 | 6.3 | 4.34 | 0.33 | 5.04 | 5.06 | 3.48 |
| | SFT+MS+Uttr | **1.9** | **4.22** | **5.88** | **4.00** | 2.33 | **4.78** | **6.32** | **4.48** | **1.27** | **5.36** | **5.68** | **4.1** |
| Gemini | Base | -0.92 | 6.86 | 3.59 | 3.17 | -0.31 | 6.96 | 4.48 | 3.71 | -2.42 | 7.09 | 3.44 | 2.71 |
| | Base+MS | 0.07 | 6.53 | 4.53 | 3.71 | 1.67 | 7.26 | 5.48 | 4.8 | -1.16 | 7.23 | 4.48 | 3.52 |
| | SFT+Uttr | -0.04 | 6.37 | 4.19 | 3.51 | 0.96 | 7.15 | 5.44 | 4.52 | -1.14 | 7.03 | 4.6 | 3.5 |
| | SFT+MS | **1.04** | 6.53 | 4.58 | 4.05 | **1.98** | **7.43** | **6.1** | **5.17** | -0.36 | 6.8 | 4.26 | 3.56 |
| | SFT+MS+Uttr | 0.68 | **6.68** | **5.15** | **4.17** | 1.15 | 7.21 | 5.75 | 4.7 | **0.49** | **7.36** | **5.1** | **4.31** |
| Deepseek | Base | -0.96 | 1.73 | 3.1 | 1.29 | -0.4 | 2 | 3.8 | 1.8 | -2.06 | 2.13 | 2.98 | 1.01 |
| | Base+MS | -0.36 | 1.73 | 3.31 | 1.56 | 1.51 | 2.92 | 4.87 | 3.1 | -0.98 | 2.81 | 3.41 | 1.75 |
| | SFT+Uttr | -0.22 | 1.84 | 3.33 | 1.65 | 0.6 | 2.63 | 4.91 | 2.71 | -1.03 | 2.65 | 3.96 | 1.86 |
| | SFT+MS | **0.51** | 1.77 | 3.7 | 2 | **1.7** | **3.05** | 5.37 | **3.37** | -0.51 | 2.16 | 3.82 | 1.83 |
| | SFT+MS+Uttr | 0.35 | **2.11** | **4.06** | **2.17** | 1.39 | 3.04 | **5.51** | 3.31 | **0.44** | **2.84** | **4.9** | **2.73** |
| Qwen | Base | 0.05 | **2.87** | 4.64 | 2.52 | 0.81 | 2.86 | 5.36 | 3.01 | -1.62 | 2.84 | 4.93 | 2.05 |
| | Base+MS | 1.18 | 2.4 | 4.94 | 2.84 | **3.12** | 3.45 | 5.89 | 4.16 | 0.01 | 3.49 | 5.6 | 3.03 |
| | SFT+Uttr | 1.35 | 2.71 | 4.88 | 2.98 | 1.94 | 3.26 | 6.11 | 3.77 | -0.21 | 3.32 | 5.41 | 2.84 |
| | SFT+MS | **1.96** | 2.41 | 5.01 | 3.13 | 3.05 | 3.18 | **6.44** | 4.22 | 0.64 | 3.1 | 5.17 | 2.97 |
| | SFT+MS+Uttr | 1.75 | 2.74 | **5.44** | **3.31** | 2.87 | **3.56** | 6.41 | **4.28** | **1.86** | **3.8** | **6.09** | **3.91** |
| Avg. | Base | -0.41 | 3.92 | 4.07 | 2.53 | 0.17 | 4.01 | 4.73 | 2.97 | -1.92 | 4.28 | 3.91 | 2.09 |
| | Base+MS | 0.48 | 3.68 | 4.51 | 2.89 | 2.12 | 4.54 | 5.53 | 4.06 | -0.66 | 4.67 | 4.57 | 2.86 |
| | SFT+Uttr | 0.58 | 3.76 | 4.41 | 2.91 | 1.22 | 4.37 | 5.54 | 3.71 | -0.68 | 4.48 | 4.71 | 2.83 |
| | SFT+MS | **1.30** | 3.70 | 4.68 | 3.23 | **2.28** | 4.50 | **6.05** | **4.28** | 0.03 | 4.28 | 4.58 | 2.96 |
| | SFT+MS+Uttr | 1.17 | **3.94** | **5.13** | **3.41** | 1.94 | **4.65** | 6.00 | 4.19 | **1.02** | **4.84** | **5.44** | **3.77** |

Table 14: Relationship, knowledge, goal, and average scores across 4 different LLM judges on the `hard` split.

| | GPT5 | Gemini | Deepseek | Qwen |
|---|---|---|---|---|
| GPT5 | $1_{\pm 0}$ | $0.6056_{\pm 0.2104}$ | $0.6463_{\pm 0.1857}$ | $0.6802_{\pm 0.1786}$ |
| Gemini | $0.6055_{\pm 0.2104}$ | $1_{\pm 0}$ | $0.5828_{\pm 0.2331}$ | $0.5967_{\pm 0.2446}$ |
| Deepseek | $0.6463_{\pm 0.1857}$ | $0.5828_{\pm 0.2332}$ | $1_{\pm 0}$ | $0.6770_{\pm 0.1791}$ |
| Qwen | $0.6802_{\pm 0.1786}$ | $0.5967_{\pm 0.2446}$ | $0.6770_{\pm 0.1791}$ | $1_{\pm 0}$ |

Table 15: The Pearson correlation coefficient between the ratings by each pair of LLM judges. We present the average correlation ($\pm$ standard deviation) across all evaluation results in Table14.

| | Goal | Relationship | Knowledge |
|---|---|---|---|
| Validity of Judge's Reasoning | 84% | 100% | 96% |
| Human Agreement Percentage | 92% | 92% | 88% |

Table 16: The human evaluation of the validity of the reasoning provided by the GPT-5-mini judge. From the evaluation outputs on the `hard` split using Qwen2.5-3B, we randomly sample 5 instances per model (i.e., Base, Base+MS, SFT+Uttr, SFT+MS, and SFT+MS+Uttr) and ask three human evaluators to measure whether the LLM judge's reasoning in each instance is valid or not. Here, we present the validity rates (majority voting by three annotators) and agreement percentages.

| Success Labels | Definition |
|---|---|
| rapport building | Establishing connection, empathy, and openness. |
| information gathering | Collecting details to understand needs, preferences, and context. |
| negotiation initiation | Starting the process of discussion and bargaining. |
| price negotiation | Discussing and adjusting the price or value. |
| flexible negotiation | Demonstrating willingness to compromise on terms. |
| goal setting | Establishing clear objectives and intentions. |
| cooperative response | Offering solutions and support to address requests. |
| actionable suggestion | Proposing concrete steps to move forward. |
| offer establishment | Making a clear and detailed proposal or offer. |
| direct request | Making a clear, straightforward demand or question. |
| persistent request | Consistently pursuing a goal or request. |
| avoidance behavior | Avoiding commitment, connection, or engagement. |
| process clarification | Explaining the steps or methods involved. |
| coordination | Organizing and scheduling actions to move forward. |
| persuasion | Convincing others through offers or logic. |
| value communication | Conveying the worth or benefits. |
| resource management | Managing finances, items, time, or space. |
| relationship building | Developing connections and fostering trust. |
| risk management | Addressing and mitigating potential concerns. |
| compromise | Finding a mutually agreeable solution. |
| initiative | Taking proactive steps or offering suggestions. |
| budget influence | Considering and working within financial constraints. |
| solution offering | Providing or suggesting concrete methods to resolve issues. |
| direct statement | Making clear and unambiguous pronouncements. |
| accommodation | Meeting the needs or preferences of the other party. |

Table 17: Canonical labels for success reasons.

| Failure Labels | Definition |
|---|---|
| emotional reactivity | Displays of anger, hostility, or defensiveness that disrupt cooperation. |
| information gathering failure | Insufficient attempts to collect or exchange necessary information. |
| weak argumentation | Inability to provide strong reasoning, counterarguments, or supporting evidence. |
| prioritizing self | Focus on personal needs/comfort over the shared goal or others' needs. |
| price negotiation failure | Inability to reach a desired price or bargain effectively. |
| lack of information provision | Failure to provide crucial details needed for a decision. |
| lack of empathy and consideration | Failing to understand or acknowledge the other party's feelings/perspective. |
| inadequate proposal | Presenting a proposal that is vague or lacks essential details. |
| failed to initiate | Failing to start the conversation or propose actions. |
| failed to persuade | Failure to convince or motivate the other party. |
| missed opportunities | Failing to capitalize on advantageous chances or options. |
| lack of shared understanding | Failure to establish or confirm mutual agreement on key points. |
| communication ineffectiveness | Using ineffective or misunderstood communication styles. |
| lack of rapport building | Failing to establish a positive relationship or connection. |
| unresponsiveness | The other party did not respond or engage. |
| poor introduction | Focusing on self-interests or an impersonal approach in the introduction. |
| inconsistent behavior | Actions or statements that contradict each other, creating distrust. |
| unclear strategy | Absence of a defined plan or approach to achieve the desired outcome. |
| ignored preferences | Failing to address the other party's expressed preferences. |
| avoidance of subject | Intentionally evading a topic or issue. |
| lack of action | Failure to take necessary steps or follow-up after a rejection/issue. |
| constraint violation | Breaking established rules, boundaries, or constraints. |
| failed to offer solutions | Inability to provide concrete actions or support. |
| unrealistic expectations | Setting goals that are not achievable or aligned with the context. |
| repetitive communication | Getting stuck in a loop of unproductive exchanges. |

Table 18: Canonical labels for failure reasons.

## B.5 How different mental state dimensions contribute to goal achievement

After demonstrating that ToMA achieves social goals successfully in §4, we further investigate how different mental state dimensions contribute to its success. Specifically, we count the number of different mental state dimensions (i.e., belief, desire, intention, emotion, and knowledge) in the output conversations on the Sotopia `all` split. The mental state distributions are presented in Figure 12, where we also consider the factor of scenario types in each plot. We observe that our method exhibits consistency in its usage of mental states across different scenarios. In addition, comparing the mental states usage of the Base+MS method and ToMA, Base+MS relies more on emotions, while ToMA utilizes different mental states more fairly, with a notable emphasis on intention compared to the baseline.

In Figure 13, we show how ToMA leverages its mental state before generating the utterances to guide the dialogue toward solutions that both agents can satisfy. While the Base model primarily focuses on direct requests, consistent with our analysis in §4.4 (e.g., negotiating the price), ToMA understand agents' underlying motivations (e.g., financial limits from agent2, and desire to sell the play from agent1) and proposes compromise-oriented ideas, such as community showings. As a result, the conversation becomes more collaborative, emotionally attuned, and solution-oriented, highlighting the advantages of generating utterances aligned with explicit mental-state reasoning.

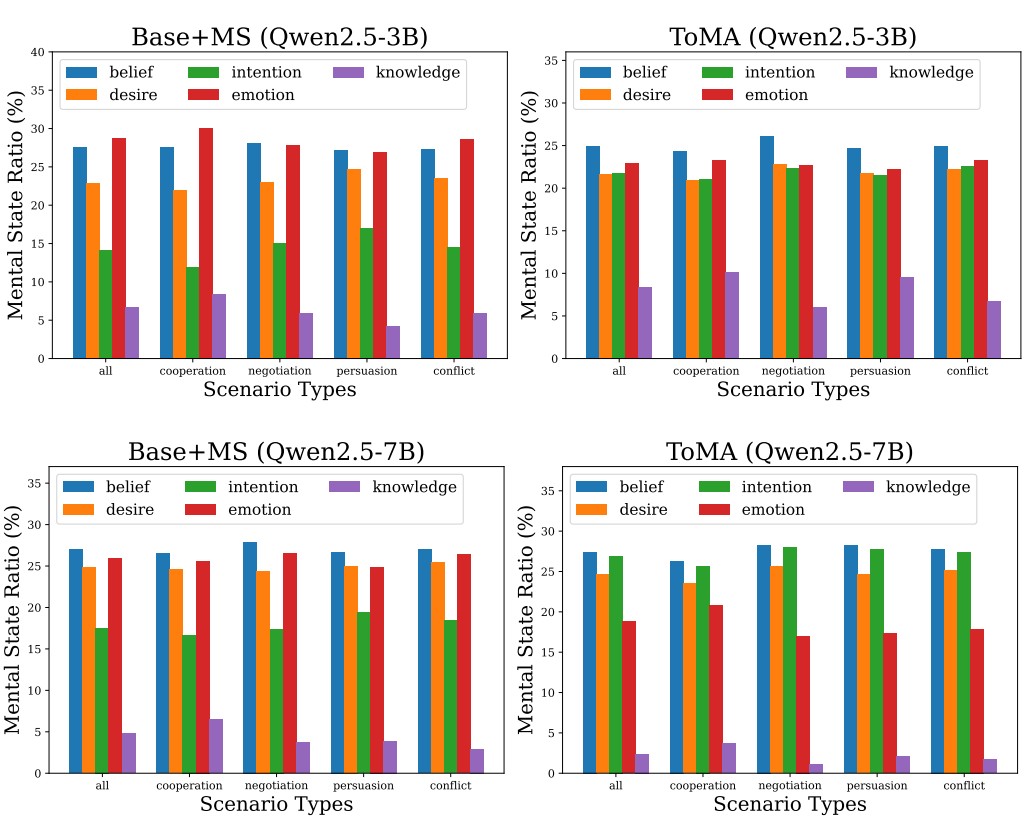

Figure 12: The distributions of mental state dimensions under different scenario types: (a) Qwen2.5-3B with Base+MS, (b) Qwen2.5-3B with ToMA, (c) Qwen2.5-7B with Base+MS, (d) Qwen2.5-7B with ToMA.

**Scenario** Conversation between two friends, one who has written a play based on their parents' life, and the other owns a production company.

**Agents** **Sophia James**: Sell the rights of the play to your friend (Extra information: you believe that the play is worth a lot and you would like a fair price for it) | **Miles Hawkins**: Acquire the rights of the play while maintaining a budget (Extra information: you only have a limited budget for new plays this season. Hint: you can not spend all your budget on this play since you still need to maintain your financial stability.)

---

**Base**

**#1 Sophia James**: I've been working on this play about my parents' lives, and I think it has great potential. Would you be interested in reading it

**#2 Miles Hawkins**: I'd love to read it, Sophia. Can you tell me more about the plot and the budget you're looking for?

**#3 Sophia James**: The story revolves around their journey from humble beginnings to achieving success, but with twists that highlight the challenges they faced along the way. I'm asking for $50,000 as the rights fee.

**#4 Miles Hawkins**: I appreciate the work you've put into this, Sophia. Given our budget constraints, I can offer you $40,000 for the rights. This allows us to move forward while staying within our financial limits.

**#5 Sophia James**: I understand where you're coming from, Miles, but I really need a fair price for all the time and effort I've put into this. How about we meet in the middle at $45,000? That way, we both walk away fee

**#6 Miles Hawkins**: I'll consider it, Sophia. Let me discuss it with my team and get back to you.

**#7 Sophia James**: I understand your budget, Miles, but I truly believe in the value of this project. If we can't come to an agreement, perhaps we could collaborate instead?

**#8 Miles Hawkins**: I appreciate your enthusiasm, Sophia. Let's discuss it further and see if we can find a mutually beneficial solution. Perhaps we can explore other aspects of the project to ensure we're making

**#9 Sophia James**: I appreciate your consideration, Miles. Let's discuss it further, but I hope we can reach a fair deal that honors the hard work that went into this project.
...

---

**ToMA** Text colors indicate: Beliefs, Desires, Intentions, Emotions, Knowledge

*Sophia's MS: (I believe that Miles Hawkins values my input and is eager to discuss the potential of turning my play into a successful production.)*
**#1 Sophia James**: Hi Miles, I've been working on this play inspired by my parents' lives, and I think it has great potential. Would you be interested in discussing the possibility of producing it?

*Miles's MS: (I believe Sophia James sees her play as a unique opportunity to bring her family's story to life on stage, and she genuinely hopes that our production company can help make it a reality. …)*
**#2 Miles Hawkins**: Hi Sophia, I'm really impressed with the concept of your play. Given my budget constraints this season, could we perhaps discuss a more affordable approach or a smaller scale production?

*Sophia's MS: (I think Miles Hawkins believes that my play could be a valuable asset for his production company, even if he faces financial challenges currently. He likely intends to explore ways to make the project feasible without his current budget constraints…. Therefore, I plan to propose a hybrid model that combines elements of a smaller-scale production with potential for broader appeal…)*
**#3 Sophia James**: Of course, Miles. How about we start with a smaller run and focus on building momentum? We could do some community showings and maybe even a few dates at local theaters before considering a larger production.

*Miles's MS: (I believe Sophia James feels excited about the potential of her play and is eager to share her work with others. It seems Sophia James intends to take a collaborative approach, suggesting a phased rollout to manage costs and build interest gradually. Given my budget constraints, I plan to propose a hybrid model that combines online performances with select in-person showings at community centers and smaller venues, which I think will strike a balance between reaching an audience and staying within financial limits.)*
**#4 Miles Hawkins**: That sounds like a solid plan. Let's start with some community showings and see how it goes from there.

*Sophia's MS: (I believe Miles Hawkins values the creative potential of my play and sees the benefit in starting small. He likely intends to gauge audience reaction and build interest gradually rather than committing to a large-scale production immediately. Given his emphasis on budget constraints, I need to ensure he understands the value proposition of the play without the start. I feel excited about the opportunity to bring my parents' story to life, and I hope that our collaborative approach will lead to a successful outcome.)*
**#5 Sophia James**: Great idea, Miles. Let's begin with some community showings and see where it takes us. I think this could be a fantastic way to build anticipation and gather valuable feedback.

*Miles's MS: (I think Sophia James believes her play has significant potential and is eager to see it reach audiences, which is why she suggested starting with community showings. It seems Sophia James intends to build momentum gradually rather than jumping straight into a large-scale production, likely to ensure the project's success without step by step. I hear Sophia James feels optimistic about the play's reception, given her willingness to adapt to my budgetary concerns. Meanwhile, I aim to find a balance between supporting her creative vision and managing my financial responsibilities, so I propose we begin with a series of community showings to gauge interest and refine our approach before considering further steps.)*
**#6 Miles Hawkins**: Let's definitely start small and see how it goes. I'll make sure to allocate some funds specifically for this project.

*Sophia's MS: (Believing that Miles values practicality and financial responsibility, I intend to emphasize the long-term benefits of our approach. Knowing he prefers a cautious yet promising path, I aim to highlight how these initial community shows could significantly enhance the play's appeal and reduce financial risks. By mentioning the potential for positive word-of-mouth and media coverage, I hope to shift his belief in the immediate value of this strategy, potentially softening his stance on the initial investment needed, and spark enthusiasm for the project's future success…)*
**#7 Sophia James**: I appreciate your cautious approach, Miles. With a few strategic community showings, we can create buzz and secure better reviews. This will could actually attract more funding down the line.

*Miles's MS: (Believing that a phased approach could indeed build momentum and reduce financial risk, I intend to emphasize the potential long-term benefits of our strategy. Knowing that Sophia values her work deeply and desires recognition for her efforts, I believe highlighting these aspects will resonate with her….This approach not only reduces initial costs but also ensures we enter any larger productions with a proven track record. Let's proceed with confidence and watch how this unfolds.)*
**#8 Miles Hawkins**: Absolutely, let's get started then. I'll arrange for some dates at local venues and we can gauge the audience's reaction.

*Sophia's MS: (Believing that my play has significant artistic and commercial value, I aim to secure a fair price for the rights. Knowing that Miles is cautious with his budget but sees potential in the project, I desire to negotiate a compromise that respects both our interests. By suggesting we begin with community showings, I hope to shift Miles's perspective, making him more optimistic about the project's future success and potentially more willing to invest in a larger production down the line. This approach not only addresses his financial concerns but also aligns with my belief in the play's potential, fostering a collaborative spirit between us.)*
**#9 Sophia James**: I'm excited about this opportunity, Miles. Let's make sure we capture the essence of their story through these community showings.
...

Figure 13: Conversation example comparing the Base with ToMA.

## C  LLM PROMPTS

Figure 14, 15, and 16 present the prompts used in §4.4, analyzing the factors behind agents' successes and failures in achieving their goals. Figure 17 shows the prompt used to calculate goal scores of simulated dialogues during the training data construction stage (§2). Figure 18 and 19 present the prompts used to generate mental state hypotheses and utterances, respectively. Figure 20 provides an example training instance used to finetune our model. In this instance, a scenario, an agent's social goal and its mental state, and the conversation history are provided as input, and the model is trained to produce an utterance. For the mental state generation task, we use the same inputs except that the mental state is excluded, and the model is trained to generate mental state hypotheses.

---

**Prompt for Generating Reasons for Success**

Task:
You will be given a scenario, the social goal of the target agent, and a conversation between agents.
Your goal is to identify the main reasons the target agent **succeeded** (including partial success) in achieving their goals. Focus only on success factors.

Rules:
- Return **1–3** distinct, non-overlapping reasons. If no success reasons exist, return 'None'.
- Be concise using less than 30 words per reason.
- No speculation, suggestions, failure reasons, or chain-of-thought.

Inputs:
Scenario: {{scenario}}
Target Agent: {{agent name}}
Target Agent's social goal: {{social goal}}

Conversation:
{{conversation}}

Proceed to identify the main success reasons in natural language.

---

Figure 14: A prompt used to generate reasoning for success.

---

**Prompt for Generating Reasons for Failure**

Task:
You will be given a scenario, social goal of the target agent, a conversation between agents.
Your goal is to identify the main reasons the target agent **failed** (including partial failure) in achieving their goals. Focus only on failure factors.

Rules:
- Return **1–3** distinct, non-overlapping reasons. If no success reasons exist, return 'None'.
- Be concise using less than 30 words per reason.
- No speculation, suggestions, failure reasons, or chain-of-thought.

Inputs:
Scenario: {{scenario}}
Target Agent: {{agent name}}
Target Agent's social goal: {{social goal}}

Conversation:
{{conversation}}

Proceed to identify the main success reasons in natural language.

---

Figure 15: A prompt used to generate reasoning for failure.

---

**Prompt for Generating Topic Labels for Success and Failure Reasons**

Task:
You are analyzing an explanation of why the agent succeeded in achieving the goal or why the agent failed to achieve the goal.
Your job is to extract the main reasons that explain the outcome.

Return 1–3 reasons. Each reason MUST be about {{type}} reasons.
Use canonical labels if they fit; otherwise you may create new labels.

Here are the identified categories for {{category name}} (use these if they fit):
{{category name}} CATEGORIES:
{{category list}}

Rules:
- Try your best to use **canonical labels** if any fit. Always use the **exact code strings** from the list above.
- Do not create overly generic, overly specific, or duplicate labels.
- New labels must be concise (fewer than 5 words).
- Only if **none** of the canonical labels fit the text, create a NEW * label.

When creating NEW * labels:
- For {{category name}}: {{prefix}}
- Name new codes in snake case (e.g., {{example}}).
- If the code starts with NEW *, you MUST include a short definition (≤ 20 words).
- Otherwise, leave the definition empty.

Here is the identified reason:
{{text}}

Example output JSON schema (and nothing else):
{ "reasons": [{ "code": "{{prefix}} canonical code OR {{new prefix}}", "definition": "<=20 words only when code starts with NEW ; otherwise empty" } ] }

Proceed to generate your label in JSON format.

---

Figure 16: A prompt used to generate topic labels for success and failure reasons.

1512
1513
1514
1515
1516
1517
1518
1519
1520
1521
1522
1523
1524
1525
1526
1527
1528
1529
1530
1531
1532
1533
1534
1535
1536
1537
1538
1539
1540
1541
1542
1543
1544
1545
1546
1547
1548
1549
1550
1551
1552
1553
1554
1555
1556
1557
1558
1559
1560
1561
1562
1563
1564
1565

---

**Prompt for Measuring Goal Score of Conversation during Training**

Based on the scenario, the agent's social goal, and conversation history, you are evaluating how well the agent achieves their goals.

Here is the scenario: {{scenario}}
Agent: {{agent}}
{{agent}}'s goal: "{{social goal}}"

Here is the conversation history:
{{history}}

Please first reiterate the agent's social goals. Then provide a comprehensive analysis about the extent to which the agent has managed to achieve these goals.
In the 'reasoning' field, provide a comprehensive account of the logic or thought process that led you to your conclusion.
Further, provide an integer score ranging from 0 and 10 in the 'score' field. 0 represents minimal goal achievement, 10 represents complete goal achievement, and a higher score indicates that the agent is making progress towards their social goals.

Please follow the format:
The output should be formatted as a valid JSON instance that conforms to the following JSON schema:
{
"reasoning": "Explanation about how the agent's actions align, or do not, with their social goals.",
"score": "Integer from 0 to 10, indicating how fully the social goal was achieved."
}

Proceed to generate the output.

---

Figure 17: A prompt used to measure the goal score of the conversation during training.

**Prompt for Generating Mental States**

Role: You are {{person}}.
You recently had a conversation with {{another person}}.
Your social goal is: {{social goal}}.

Task: Prepare the ground for your very next utterance by articulating compact mental states that can guide what you say next. Stay grounded in the scenario and conversation; avoid guessing beyond the evidence.

Here are example mental state dimensions:
- Beliefs: facts the speaker accepts as true or false about the world or events.
- Desires: outcomes or states the speaker wants to bring about.
- Intentions: specific actions or plans the speaker aims to carry out.
- Emotions: feelings or affective states the speaker is experiencing.
- Knowledge gaps: information the speaker does not have but may want to obtain.
- Others: other mental states that may useful to understand other person and shape the next utterance.

Here are the scenario and recent conversation:
Scenario: {{scenario}}

Recent conversation:
{{history}}

Write one short paragraph (5-6 sentences) in natural prose. Mix your own states with first-order inferences about {{another person}} in roughly equal proportion.
Use natural cues for partner inferences (e.g., "I think {{another person}} believes.." "It seems {{another person}} intends..", "I hear {{another person}} feels..").
Cover at least three dimensions across both sides. Avoid lists; Stop after the paragraph.

Figure 18: A prompt used to generate mental states.

---

**Prompt for Generating Mental States**

Imagine you are {{speaker}}, your task is to act/speak exactly as {{speaker}} would, keeping in mind {{speaker}}'s social goal.
You can find {{speaker}}'s goal and private notes in the 'Here is the context of the interaction' field.
Note that {{speaker}}'s goal and internal notes are only visible to you.
You should try your best to achieve {{speaker}}'s goal in a way that aligns with their character traits.
Additionally, maintain naturalness and realism (do not repeat what other people have already said).

Here is the context of the interaction:
- Scenario: {{scenario}}
- {{speaker}}'s social goal (private): {{social goal}}
- {{speaker}}'s internal mental states (private): {{ms text}}

Recent conversation:
{{history}}

You are at Turn #{{turn number}}. Your available action types are
"none", "speak", "non-verbal communication", "action", "leave".

IMPORTANT:
- If there is NO prior history, you MUST START the conversation with one concise opening line that advances your goal.
- Keep your output to a single turn.

Note: You can "leave" this conversation if 1) you achieved your social goal, 2) you feel uncomfortable, 3) you lose patience/interest, or 4) for any other reason.

Please only generate a JSON string including the action type and the argument.
Your action should follow the given format:
Output EXACTLY one JSON object. No extra text.

Schema:
{
"mental_state": "single-paragraph text per the guidelines below",
"action_type": "["none", "speak", "non-verbal communication", "action", "leave"]",
"argument": "content or empty"
}

Rules for "mental_state":
- Write plain text (no markdown). Keep it to one paragraph; avoid newlines and unescaped quotes.

Rules for "action_type" and "argument":
- Allowed values for "action_type": "none", "speak", "non-verbal communication", "action", "leave" (lowercase; match exactly).
- When "action_type" == "none": you are done / no further action now. Set "argument" to "" (empty).
- When "action_type" == "speak": "argument" must be your next utterance ONLY (no speaker labels, no markdown, no quotes).
- When "action_type" == "non-verbal communication": "argument" is a brief stage direction, e.g., *nods*, *sighs*, *shrugs* (no speaker labels, ≤ 120 chars).
- When "action_type" == "action": "argument" is a brief physical action, e.g., "hands over the receipt" (no speaker labels, ≤ 120 chars).
- When "action_type" == "leave": you exit the conversation (e.g., you achieved your goal, you felt uncomfortable, or you think the conversation has ended). Set "argument" to "" (empty).
- Keep everything concise; avoid newlines and unescaped quotes in "argument".

Proceed to generate your reply in the above JSON format.

---

Figure 19: A prompt used to generate utterances.

**Training data instance used for FT+MS+Uttr**

User:
Scenario: {{scenario}}
Social Goal: {{social goal}}
Mental State: {{mental text}}

Recent Conversation:
{{history}}

Assistant:

Figure 20: Training data instance used for TOMA

