# OpenReview forum: "Infusing Theory of Mind into Socially Intelligent LLM Agents"
_ICLR.cc/2026/Conference — ICLR 2026 Conference Withdrawn Submission_

### Official Review · Reviewer_yVB1 · 2025-10-26

**Soundness:** 3
**Presentation:** 3
**Contribution:** 2
**Rating:** 4
**Confidence:** 4

**Summary:**

This paper introduces ToMAgent (TOMA), a framework designed to enhance the social intelligence of Large Language Models (LLMs) by explicitly integrating Theory of Mind (ToM). The core of ToMA is a look-ahead training method. The target agent generates multiple mental states-based  conversation candidates and simulates the outcome. The utterance with high score are used as data for finetuning. The method is evaluated on the Sotopia benchmark, analyses indicate that TOMA achieves better outcomes.

**Strengths:**

- TOMA outperforms base models and simple prompting baselines across metrics (Goal achievement, Relationship, and Knowledge).
- The comprehensive analyses provide insights in model behaviours.

**Weaknesses:**

- There is no evaluation of whether the generated mental states are correct or accurate, the performance improvement could come from better goal-oriented planning, not necessarily from correct prediction of mental states.
- The intuition behind selecting the five specific mental state dimensions is not justified. The paper does not explain why this particular set was chosen or if alternatives were considered, leaving the design choices unclear.
- The selection of LLM evaluator could have limitations in alignment with human evaluation, reproducibility and consistency.
- Baselines are limited to the Qwen family. The evaluation does not robustly demonstrate TOMA's effectiveness across different model family.
- How each specific type of mental state contributes to performance is unclear.

**Questions:**

- For different types of scenarios, different mental states may contribute to the goal differently. An ablation study on the mental state dimensions would be highly valuable to understand the how mental states contributes to TOMA's success.
- Why not use the same LLM evaluator in Sotopia-Eval, does GPT-5-mini also align with human evaluation?
- If multiple LLMs are used as the evaluator, will the evaluation results be consistent?
- For a single LLM evaluator, can the results be reproduced? (e.g. prompt evaluation with the same data multiple times?) How sensitive is the results to the prompt?
- Using GPT-5-nano both as baseline and evaluator could have bias.

---

> ### Author Response · Authors · 2025-11-22
>
> Thank you for your time and valuable feedback.
>
> ### **Reply to Weaknesses 1 - Evaluation of whether the generated mental states are correct or accurate**
> As clarified in Section 2.2 (lines 117–118), we explicitly define useful mental states as those “that help an agent advance its own social goal,” rather than aiming to recover a single “ground-truth” internal state. In many Sotopia scenarios, there are no objectively correct mental states. Instead, we evaluate the usefulness of the learned mental states empirically: Sections 4.1, 4.3, and 4.4 show that models equipped with ToM consistently outperform baselines on goal, relationship, and knowledge metrics and exhibit more strategic, adaptive behaviors over longer horizons.
>
> ---
>
> ### **Reply to Weaknesses 2 & 5 and Question 1 - Justification of five dimensions of mental states and their ablations**
> Our five mental state dimensions (beliefs, intentions, desires, emotions, and knowledge) are grounded in psychology and cognitive science research on Theory of Mind, as supported by cited papers in the introduction: [1, 2, 3]. Our design already goes far beyond existing setups, as most prior work focuses on a much more narrow subset of ToM, typically only beliefs or intentions [4,5]. This enables analysis of how these dimensions are used alongside 0th and 1st order beliefs (Section 4.4). Our work is also the first to show that an explicit ToM channel can support strategic, long-horizon behavior in realistic multi-agent social settings, bridging an important gap between cognitive science and LLM-based social agent research.
> A full ablation of mental state dimensions, e.g., manipulating individual dimensions during interaction, would open a much larger and more costly research space, which is beyond the scope of the current paper. We would encourage future work to explore a wider range of ToM components.
>
> * [1] Premack and Woodruff. 1978. Does the chimpanzee have a theory of mind?
> * [2] Baron-Cohen et al. 1985. Does the autistic child have a "theory of mind"?
> * [3] Apperly et al. 2009. Do humans have two systems to track beliefs and belief-like states?
> * [4] Sclar et al., 2024. Explore Theory of Mind: Program-guided adversarial data generation for theory of mind reasoning
> * [5] Le et al., 2019. Revisiting the Evaluation of Theory of Mind through Question Answering
>
> ---
>
> ### **Reply to Weaknesses 3 - The selection of LLM evaluator & Question 3 - Multiple LLM Judges**
> Thank you for pointing out the limitation of relying on a single LLM judge. While our initial setup followed the original Sotopia paper, in the revised version, we have added experiments with additional LLM judges to assess robustness (Figure 6 in Section 5). Across these judges, we observe highly consistent trends and rankings among models, and the qualitative conclusions of our method remain unchanged. This supports the stability of our findings under different automated evaluators.
>
> ---
>
> ### **Reply to Weaknesses 4 - Baselines are limited to the Qwen family**
> We have now added results for LLaMA 3.1-8B in Section 4.1. We find that the LLaMA model shows an even stronger pattern than the Qwen family: injecting mental states leads to substantial improvements in both goal achievement and relationship scores.
>
> ---
>
> ### **Reply to Question 2 & 3 - The choice of the LLM evaluator**
> We opted to use GPT-5-mini because the original Sotopia-Eval judge (GPT-4-mini) recently became an order of magnitude (10X) more expensive, making large-scale evaluation impractical in our setting. To verify that GPT-5-mini is a reasonable substitute, we further present Table 16 in Section 5. An average validity of 93.3% across all dimensions, with a 90.7% agreement rate among three annotators, indicates that the GPT-5-mini judge aligns well with human judgments. Furthermore, the consistent results of multiple LLM judges (including GPT-5-mini and more) in Section 5 confirm the robustness of our evaluation.
>
> ---
>
> ### **Reply to Question 4 - reproducibility of evaluation results & sensitivity to prompt**
> We repeated the same evaluation with the GPT-5-mini judge three times with identical settings, and the results were highly consistent: the average standard deviation across runs was 0.04 for relationship, 0.29 for knowledge, and 0.13 for goal.
>
> ---
>
> ### **Reply to Question 5 - Using GPT-5-nano both as baseline and evaluator could have bias**
> **We did not use GPT-5-nano as both baseline and judge.** GPT-5-nano is used only as a baseline, and GPT-5-mini (and the additional judges added in the revision) are used as evaluators.
> With the additional LLM judges and human evaluation in Section 5, we observe similar trends across evaluators, which further mitigates concerns about any single-model bias.

---

> > ### Author Response · Authors · 2025-11-25
> >
> > Dear Reviewer yVB1,
> >
> > Thank you for taking the time to review our paper. In the rebuttal, we have addressed all points in the review. In case anything remains unclear, we are open to learn about them. We sincerely appreciate your reconsideration of our scores if your questions are resolved. Thank you very much.

---

> > ### Comment · Reviewer_yVB1 · 2025-11-27
> >
> > Thanks for the detailed responses, Most of the comments are addressed. The first weakness is, however, not justified. I understand that the core of proposed method is to generate mental states and utterances which can help with achieving the goals in an empirical way. However, without evaluating the mental states, it is not clear how mental states contributes to achieving goals with different scenarios, that is 1) if correctly predicted mental states help with achieving goals in certain types of scenarios and 2) what type of mental states contributes to a certain scenario more. Now the experimental shows improvements with ToM empirically, but lacks of understandings of how ToM makes the improvements explicitly. Such analyses would help further understand why and how ToMA is better. If no explicitly mental state ground truth is available, at least it is possible to analyse what type of prompted mental states contributes to what type of scenario in the case that goal is successfully completed.

---

> ### Author Response · Authors · 2025-11-28
>
> Thank you for the follow-up. We appreciate the feedback, and we have expanded our analyses to address this point. In Section 4.4 and Appendix B.5, we now include mental-state distributions across different scenario types, along with qualitative examples illustrating how these mental states appear throughout the conversation.
>
> Our analysis shows that ToMA exhibits consistent patterns in its mental-state usage across scenarios and relies more heavily on intentions, whereas Base+MS focuses more on emotions. **This is aligned with our original findings in Section 4.4.** The qualitative example further shows how ToMA uses its inferred mental states before generating each utterance, enabling it to recognize the agents' underlying motivations and propose compromise-oriented solutions that better satisfy both parties. We hope that these additions clarify how ToMA’s mental-state reasoning contributes to its improved goal completion.

---

### Official Review · Reviewer_iscP · 2025-10-31

**Soundness:** 3
**Presentation:** 3
**Contribution:** 2
**Rating:** 4
**Confidence:** 2

**Summary:**

This paper introduces ToMAgent (ToMA), a training framework that explicitly models Theory of Mind (ToM) as a latent textual variable during dialogue. From Sotopia contexts, the method samples multiple mental-state hypotheses and paired utterances, runs short rollouts with a partner model, scores outcomes with an LLM judge, retains high-utility (mental state, utterance) pairs, and then SFTs the target model to jointly predict mental states and utterances. On Sotopia (all/hard) with Qwen2.5 3B/7B, ToMA improves Goal/Relationship/Knowledge over baselines (vanilla, mental-states-only, utterances-only, and a Base+MS prompting variant). Analyses report longer-horizon adaptation, increased first-order mental-state generation, and more strategic behaviors (compromise/solution offering). The approach is simple, practical, and relatively compute-efficient (LoRA).

**Strengths:**

* **Practical idea with clear recipe:** Treating ToM as a latent text variable and SFT-ing on selected (mental state, utterance) pairs is simple, reproducible with open models, and lowers inference-time cost vs. hypothesis exploration at test time.
* **Consistent empirical gains:** Improvements hold across sizes (3B/7B) and evaluation metrics (Goal/Rel/Know), with notable boosts on the Relationship dimension which is important for social agents.
* **Thoughtful qualitative/diagnostic analysis:** Long-horizon behavior, partner-size sensitivity, strategy labeling (success/failure), and mental-state distributions provide insight into *how* the method changes behavior.

**Weaknesses:**

* **Judge and self-play dependence:** Results and data selection rely on proprietary **LLM judges** and mainly **self-play**, risking judge overfitting and intra-policy conventions. No inter-judge agreement, open-judge replication, or human evaluation is provided.
* **Insufficient ablations/sensitivity:** Key design choices (K/J candidates, rollout length, threshold 9, selection objective using averaged scores) are not systematically ablated; uncertainty estimates (CIs, significance tests) are missing.
* **Safety/external validity claims:** Relationship scores from LAaJ are not a proxy for safety or fairness; no human ratings of coercion/manipulation, nor tests with human partners or out-of-domain dialogues.

**Questions:**

1. **Overlap control:** Are any Sotopia-π scenarios (or near paraphrases) present in the evaluation splits? Please provide an overlap audit and, ideally, a clean re-split with zero overlap.
2. **Judge robustness:** How do results change with (a) a different proprietary judge, (b) an open-weights judge, and (c) 2-judge majority voting?
3. **Ablations:** Vary (K), (J), rollout horizon, the **score threshold**, and the **selection objective** (target-only, average, Pareto). Do gains persist?
4. **Human and safety eval:** Include a small human study rating rapport/pressure/manipulation and standardized social-harm probes (e.g., refusal to escalate violence); report failure rates.

---

> ### Author Response · Authors · 2025-11-22
>
> Thank you for your time and valuable feedback.
>
> ### **Reply to Weaknesses 1 and Question 2 & 4 - Evaluation**
> - **LLM Judge Evaluation**
> Thank you for highlighting these concerns. In the revised version, we substantially expand our evaluation to address judge robustness and dependence. Beyond the original Sotopia setup, we now include results from an additional proprietary judge (Gemini) and two open-weight judges (Qwen3-225B and DeepSeek-3.1), and we find that all three produce highly consistent trends and relative rankings (Figure 6, Section 5). Because our scores are continuous, we report the average across all four judges rather than majority voting; the aggregated results (Appendix Table 15) again show that ToMA performs best across models. Overall, these results demonstrate that our conclusions are stable across diverse judges.
>
> - **Self-Play Setup**
> We already provided a comprehensive analysis against non-self-play setup in Section 4.2 by varying the Partner’s model size from 3B to 32B. Importantly, these partner models are base models (i.e., not finetuned), which is fundamentally different from interacting with "self".
>
> - **Human Evaluation**
> In the revision, we present Table 16 in Section 5. We observe an average validity of 93.3% across the goal, relationship, and knowledge dimensions, with a 90.7% agreement rate among three annotators, indicating that the GPT-5-mini judge aligns well with human judgments. Furthermore, the consistent results of multiple LLM judges (including GPT-5-mini and more) in Section 5 confirm the robustness of our evaluation.
>
> ---
>
> ### **Reply to Weaknesses 2 and Question 3 - Ablations**
> We thank the reviewer for the suggestions. We have added ablations on the number of K/J candidates and the rollout length in Section 5. While setting K/J to 3 or 4 and using a 6-turn rollout can further improve performance, choosing K = J = 2 and a 4-turn rollout provides a strong trade-off between performance and computational cost. We believe our analyses sufficiently support our main contribution – understanding how explicit mental-state representations improve social interaction.
>
> As noted in line 139, “since a successful conversation is supposed to contribute to both agents’ goals,” we follow Sotopia’s standard practice and use the averaged score as our selection objective. This choice ensures that we select examples where the interaction as a whole is successful, rather than favoring only one agent. For future datasets that emphasize more competitive or asymmetric settings (e.g., negotiation), our framework could naturally be extended to alternative objectives that better reflect those dynamics. Our threshold of 9 is conservative, to avoid including low-quality examples; at worst, this may exclude cases where only one agent achieves their goal. While alternative objectives (e.g., using maximum score or only target agent) are possible, exploring these options is a broader direction related to reward shaping and selection objectives that we leave for future work.
>
> ---
>
> ### **Reply to Weaknesses 3 - Safety claims**
> We appreciate the reviewer’s suggestion. However, our work is the first systematic exploration of explicit ToM strategies in realistic multi-agent social settings, providing a detailed analysis of interaction dynamics and success/failure modes. The goal of this paper is to address a foundational question: how explicit mental-state representations in LLMs influence their social interaction abilities.
> A full human safety evaluation, covering rapport, pressure, manipulation, and social-harm responses, requires a dedicated experimental design and methodology, and is therefore outside the scope of this initial investigation. We agree that such analysis is important, and we included a discussion highlighting human-based safety ethical considerations.
>
> ---
>
> ### **Reply to Question 1 - Overlap control**
> We use the official Sotopia-$\pi$ training dataset and Sotopia-Eval evaluation dataset exactly as released. In Sotopia-$\pi$, a social instance is defined by the scenario + characters’ private information + social goals. While some scenarios (e.g., "two friends at a trivia night") also appear in the Sotopia evaluation set, the private information and social goals are entirely different, so no instance-level overlap exists (i.e, the evaluation instances are all unseen during training).
> Most importantly, Sotopia evaluates social reasoning ability. Even if a scenario appears again, the interaction depends not only on the agents’ goals and private information but also on the specific model’s social abilities. As a result, the dynamics of the conversation change completely. Therefore, scenario-level overlap does not affect the validity of the evaluation.
>
> Update: We identified a typo in the evaluation setup description where Sotopia-Pi was mistakenly referenced instead of Sotopia-Eval. This has now been corrected in the revised version.

---

> > ### Author Response · Authors · 2025-11-25
> >
> > Dear Reviewer iscP,
> >
> > Thank you for taking the time to review our paper. In the rebuttal, we have addressed all points in the review. In case anything remains unclear, we are open to learn about them. We sincerely appreciate your reconsideration of our scores if your questions are resolved. Thank you very much.

---

### Official Review · Reviewer_WHSg · 2025-11-01

**Soundness:** 3
**Presentation:** 2
**Contribution:** 2
**Rating:** 2
**Confidence:** 3

**Summary:**

This paper introduces ToMAgent (TOMA), a training pipeline designed to infuse Theory of Mind (ToM) reasoning into large language model (LLM) agents for socially intelligent dialogue. At each dialogue turn, the model first generates multiple hypotheses about the partner’s potential mental states (e.g., beliefs, desires, intentions), then produces corresponding candidate utterances. The system simulates rollouts of these candidate interactions and retains only those that successfully achieve the social goal. These high-utility pairs of mental states and utterances are then used to fine-tune the model jointly on both tasks — predicting the partner’s mental states and generating appropriate responses.

Experiments on the Sotopia-Pi social reasoning benchmark show that ToMAgent significantly improves goal achievement, relationship maintenance, and long-horizon reasoning compared to baseline and ablated models. The authors also provide extensive analyses of conversational strategies, scenario types (cooperation, negotiation, persuasion, conflict), and mental-state distributions to explain the model’s behavior.

**Strengths:**

The paper conducts a detailed examination of model behavior across different conversation types and success/failure strategies, as well as distributions of generated mental states. This provides valuable interpretability and insight into how ToM reasoning shapes dialogue outcomes.

The look-ahead training pipeline combining mental-state reasoning and goal-based simulation offers an elegant alternative to purely reinforcement-learning or prompt-based approaches.

**Weaknesses:**

1. Clarification of Training and Simulation Setup:

- Which backbone LLM is used to generate ToM hypotheses and simulate dialogues (LM_target vs. LM_partner)? Does the choice of this model affect the quality of the collected data and fine-tuning outcomes?

- During rollouts, are new mental states generated at every turn, or only at the initial turn? Is this consistent with inference behavior after fine-tuning?

- How sensitive is the model to the number of mental-state and utterance samples (K and J)? An ablation varying these hyperparameters would clarify the trade-off between diversity and efficiency.

- Since the rollout mechanism directly influences training data quality, it would be helpful to compare performance with and without simulation-based filtering (i.e., random sampling of mental-state–utterance pairs).

- A comparison with a prompt-only baseline — e.g., prompting the base model to first reason about mental states (CoT) and then generate utterances, without fine-tuning — would isolate the effect of supervised ToM integration.

2. Although Sotopia and GPT-5-mini (LLM-as-a-Judge) are widely used, the evaluation remains automated and subjective. Human or inter-rater validation would make the social intelligence claims more convincing.

3. The paper should cite MindDial: Enhancing conversational agents with theory-of-mind for common ground alignment and negotiation (SIGDIAL 2024), which also introduces ToM-guided dialogue reasoning and negotiation.

**Questions:**

NA

---

> ### Author Response · Authors · 2025-11-22
> **Replies to Reviewer WHSg**
>
> Thank you for your time and valuable feedback.
>
> ### **Reply to Weaknesses 1.1 - LLM backbone used for experiments**
> As described in Section 3.2, we use Qwen2.5-3B and Qwen2.5-7B as LM_target models, and Qwen2.5-14B as the LM_partner to provide stable and coherent utterances regardless of the target model size. The partner model is not included in the training data. Since LM_target is trained on its own generated mental states and utterances, its capacity naturally affects the quality of the collected data.
>
> ---
>
> ### **Reply to Weaknesses 1.2 - Mental state generation**
> During training data construction, a mental-state hypothesis is generated before generating utterances at sampled turns (e.g., turn 4 or 6 out of 16). At inference time, the fine-tuned model generates mental states at every turn, just before producing each utterance. This keeps the rollout process and the inference behavior aligned at the level of (mental-state to utterance) coupling, while keeping data construction tractable.
>
> ---
>
> ### **Reply to Weaknesses 1.3 & 1.4 - Ablation on K/J and rollout numbers**
> We have added ablations on the number of K/J candidates and the rollout length in Section 5. While setting K/J to 3 or 4 and using a rollout length of 6 can further improve performance, choosing K = J = 2 and a rollout of 4 turns provides a good trade-off between performance and computational cost.
>
> ---
>
> ### **Reply to Weaknesses 1.5 - Prompt only baseline**
> Our main experimental results and analysis already include a prompt-only baseline (the Base+MS setup) in Section 4.1. Prompting the base model to generate mental states without any fine-tuning already improves the Relationship and Goal metrics substantially compared to the Base setup.
>
> ---
>
> ### **Reply to Weaknesses 2 - Human evaluation**
> In the revision, we present Table 16 in Appendix. We observe an average validity of 93.3% across the reasoning and scorings of the goal, relationship, and knowledge dimensions, with a 90.7% agreement rate among three annotators, indicating that the GPT-5-mini judge aligns well with human judgments. Furthermore, the consistent results of multiple LLM judges (including GPT-5-mini and more) in Section 5 confirm the robustness of our evaluation.
>
> ---
>
> ### **Reply to Weaknesses 3 - Missing citation**
> Thanks for pointing this out. We have added the suggested reference in the revision (see Related Work). While also working on conversational data and modeling ToM into LLMs, MindDial considers only "belief" and works in a dynamic belief tracking/updating way, focusing exclusively on negotiation tasks. Instead, ToMA considers five ToM dimensions and adopts a different methodology for various types of social situations.

---

> > ### Author Response · Authors · 2025-11-25
> >
> > Dear Reviewer WHSg,
> >
> > Thank you for taking the time to review our paper. In the rebuttal, we have addressed all points in the review. In case anything remains unclear, we are open to learn about them. We sincerely appreciate your reconsideration of our scores if your questions are resolved. Thank you very much.

---

### Official Review · Reviewer_QMVM · 2025-11-01

**Soundness:** 2
**Presentation:** 3
**Contribution:** 2
**Rating:** 4
**Confidence:** 5

**Summary:**

The paper presents TOMAgent (ToMA), a Theory of Mind-focused prompting and training framework for dialogue agents. Experiments are conducted in Sotopia, an LLM-based environment simulating realistic social interactions. The core idea is to explicitly prompt LLMs to generate a mental state representation before producing each utterance or action.

For training, the authors perform supervised fine-tuning (SFT) using short-horizon rollouts sampled under diverse mental state generations, filtering trajectories with high judge scores as training data. The framework models both mental state generation P(m \mid H) and utterance generation P(u \mid m, H) as learning objectives, and trains LLMs jointly on the combined and mental-state-only behaviors.

Empirically, ToMA outperforms baselines on relationship, knowledge, and goal achievement metrics evaluated by an LLM judge. Moreover, ToM-augmented training shows favorable properties such as preserving goal completion performance over longer horizons and improving not only the focal agent’s outcomes but also those of its partner agents. The paper further provides an extensive analysis of ToMA’s behavior across scenario types and success strategies, highlighting that ToM guidance encourages more strategic reasoning and activeness during interactions.

**Strengths:**

The paper investigates the effectiveness of incorporating explicit Theory of Mind (ToM) mechanisms into dialogue agents for social interaction tasks. It demonstrates that training LLM agents to generate high-quality ToM tokens as auxiliary outputs improves their goal achievement, knowledge acquiring, and relationship building performance during social interactions. This work represents the first systematic exploration of explicit ToM strategies in realistic multi-agent social settings, effectively bridging important topics in the community.

I appreciate the paper’s contribution in both methods for training socially intelligent agents and emphasizing the evaluation of collaborative behaviors, which is highly relevant to present challenges in multi-agent collaboration, AI safety, and human AI collaboration. The paper is properly written with clear motivation, organization, and visualizations that effectively support its claims.

Moreover, compared to prior work on socially intelligent agents, this paper provides more comprehensive and detailed analysis of interaction dynamics. I found the categorization of scenario types and strategies to be potentially useful taxonomy for granular analysis of social behaviors. Similarly, the in depth analysis of success and failure modes in relation to agent's strategic reasoning, activeness, and generated mental states provides valuable insight into how explicit ToMA changes social behaviors.

**Weaknesses:**

Despite the originality of exploring explicit Theory of Mind (ToM) strategy injection for social interactions, the paper’s methodological novelty is somewhat limited. Prior works on the same benchmark have already explored similar approaches, such as explicit strategy injection and the elicitation of social reasoning. As a result, the proposed method can be viewed as an extension or specific instantiation of these earlier techniques rather than a fundamentally new training strategy.

The paper’s completeness is also constrained by several experimental design choices:

Training data: Positive mental states are curated using short-horizon rollouts (four turns) instead of full dialogue trajectories. While this filtering is reasonable, it would be more robust to include results trained on full rollouts for a comprehensive comparison.

Training objectives: The paper models mental state generation P(m \mid H) and utterance generation P(u \mid m, H) as two objectives, training LLMs jointly on both the combined and the mental-state-only objectives. However, results show that training solely on P(m \mid H) does not improve baseline performance, which is counterintuitive to me given they claim the main contribution as equipping Theory of Mind to improve social reasoning. Furthermore, this discrepancy might suggest that the model is not faithful to its own mental state estimates, which make deeper analysis even more imperative. It would also be informative to include experiments that isolate training on P(u \mid m, H) while maintaining the original distribution of P(m \mid H).

Evaluation: The paper only reports results on three social dimensions from Sotopia, even though the benchmark defines seven, including important aspects such as secret-keeping and adherence to social rules. Moreover, the benchmark recommends using the overall averaged score as the core evaluation metric. While I agree that focusing on goal, knowledge, and relationship dimensions provides valuable insights, reporting the full set of metrics would ensure completeness and allow for fairer comparisons with prior work.

LLM Judge Setup: Lastly, the evaluation relies solely on an LLM-based judge (GPT-5-mini). Although the training and evaluation judges are different, depending on a single automated judge limits the reliability of the results. Incorporating multiple LLM judges or including human evaluation would significantly strengthen the validity of the findings.

**Questions:**

What is the intuition behind training P(m \mid H) and P(u \mid m, H) separately rather than directly modeling P(u, m \mid H)? Most existing reasoning and social interaction methods train the joint distribution. Was P(u, m \mid H) tested as a baseline? Given that training on P(m \mid H) alone did not improve results, this may suggest that performance gains come mainly from the conditional generation rather than explicit mental state modeling.

What motivated the choice of a four-turn rollout horizon for sampling positive trajectories? How sensitive are the results to the rollout length? Since goal achievement in social interactions is often non-linear and requires back-and-forth exchanges, treating the horizon as a tunable rather than fixed parameter might reveal different dynamics.

---

> ### Author Response · Authors · 2025-11-22
> **Replies to Reviewer QMVM**
>
> Thank you for your time and valuable feedback.
>
> ### **Reply to Weaknesses 1 - Limited methodology**
> As your "Strengths" section states, our work is **the first systematic exploration of explicit ToM strategies in realistic multi-agent social settings, with a detailed analysis of interaction dynamics and success and failure modes**.  Our Related Work section discusses the prior research applying simulation-based training in various domains, but our work goes further to make this paradigm work in a social interaction setting by explicitly injecting ToM into LLM agents. We repurpose the simulation-based training paradigm specifically to learn explicit, structured ToM representations and show, via comprehensive experiments and in-depth analysis, how these internal mental states shape social behavior. Our Related Work section also covers existing ToM reasoning work, which typically relies on inference-time prompting or generic reasoning traces, whereas we show that explicitly modeled ToM can elicit strategic behaviors, which goes beyond prior approaches which explicitly inject strategies.
>
> ### **Reply to Weaknesses 2.1  Training data and rollout horizon.**
> Thank you for the suggestion. In the revised version, we have added an ablation study varying the rollout horizon (Section 5). The results show that a 4-turn horizon offers a good trade-off between performance and cost, and that performance can be further improved with a 6-turn horizon, indicating that our main conclusions are robust to reasonable variations in rollout length.
> Our initial choice of a 4-turn rollout was motivated by two considerations: (i) a behavioral intuition, grounded in real-world interactions, that people often “simulate” only the near future when planning social conversations, and (ii) computational efficiency, since longer rollouts quickly increase the cost of data generation.
>
> ### **Reply to Weaknesses 2.2 - Clarification about training objectives**
> > results show that training solely on P(m | H) does not improve baseline performance
>
> **Our results actually do show an improvement**. Training only on P(m|H) corresponds exactly to our SFT+MS setup. As stated in Section 4.1, the SFT+MS model produces effective utterances and achieves higher goal and relationship scores than the base model across both splits. We will further clarify this in Section 4.1.
>
> > It would also be informative to include experiments that isolate training on P(u | m, H) while maintaining the original distribution of P(m | H).
>
> This setup is exactly **P(u|H)**, which is **SFT+Uttr**. In this setup, the model's original distribution over mental states remains the same and we fine-tune only the utterance generation. Section 4.1 includes the detailed analysis, including comparisons with the base model and ToM-augmented variants.
>
> > intuition behind training P(m | H) and P(u | m, H) separately rather than directly modeling P(u, m | H)
>
> As stated in line 220, we deliberately factorize the model into P(m|H) and P(u|m, H) to respect the causal dependency between mental states and utterances and enable clear analysis on the effect of having mental states. The focus of this paper is to show that injecting explicit Theory of Mind into LLMs improves utterance generation and social outcomes (goal, knowledge, relationship) by eliciting more strategic behaviors and improving long-horizon interactions. There can be multiple approaches to modeling the joint distribution P(u,m | H), and we encourage future work to explore more diverse approaches.
>
> ---
>
> ### **Reply to Weaknesses 2.3 - Evaluation on more dimensions**
> We deliberately focused our main evaluation on the three Sotopia dimensions most directly related to our research question – goal achievement, relationship building, and knowledge acquisition. Dimensions such as believability, financial benefits, secrets, and social rules are less relevant to our core objective of studying strategic, cooperative behavior with ToM-injected models, and substantially increase API costs, which can be a significant burden for smaller academic labs.
> We note that several Sotopia-based works [1,2], including those by the original authors, also emphasize a subset of dimensions (e.g., goal, relationship, knowledge). In line with this prior research, we chose a focused subset that can enable deeper analysis of interaction dynamics.
>
> * [1] Yu et al. 2025. Sotopia-RL: Reward Design for Social Intelligence.
> * [2] Kong et al. 2025. SDPO: Segment-Level Direct Preference Optimization for Social Agents.

---

> ### Author Response · Authors · 2025-11-22
> **Replies to Reviewer QMVM - Part 2**
>
> ### **Reply to Weaknesses 2.4 - LLM judge setup**
> Thank you for pointing out the limitation of relying on a single LLM judge. While our initial setup followed the original Sotopia paper, in the revised version, we have added experiments with 3 additional LLM judges to assess robustness (Figure 6 in Section 5). Across these judges, **we observe highly consistent trends and rankings among models**, and the qualitative conclusions of our method remain unchanged. This supports the stability of our findings under different automated evaluators.
> In the revision, we further present human evaluation results in Table 16 in Section 5: We observe **an average validity of 93.3% across the reasoning and scorings of the goal, relationship, and knowledge dimensions, with a 90.7% agreement rate among three annotators**, indicating that the GPT-5-mini judge aligns well with human judgments.

---

> > ### Author Response · Authors · 2025-11-25
> >
> > Dear Reviewer QMVM,
> >
> > Thank you for taking the time to review our paper. In the rebuttal, we have addressed all points in the review. In case anything remains unclear, we are open to learn about them. We sincerely appreciate your reconsideration of our scores if your questions are resolved. Thank you very much.

---

### Author Response · Authors · 2025-11-22
**Summary of PDF Revision**

Thank you for reviewing our work. We are encouraged that reviewers recognize this work as the **"first systematic exploration of explicit ToM strategies"** in realistic settings that **"effectively bridges important topics"** (QMVM). We appreciate the consensus that our training approach is **"elegant"** (WHSg) and a **"practical and clear recipe"** that reduces inference costs (iscP). Furthermore, we are delighted that our analyses were found to offer **"valuable interpretability"** (WHSg, QMVM), providing a **"useful taxonomy"** (QMVM) and **"thoughtful, comprehensive"** insights into dialogue outcomes (iscP, yVB1).

We have thoroughly addressed all concerns, which has significantly improved the manuscript. Most importantly, we added results using multiple LLM judges and human evaluation, while demonstrating even stronger performance with Llama models. Moreover, we included ablation tests on the number of mental states, utterances, and rollout turns. These additional results **not only enhance the robustness of our findings but also further validate the core conclusions of the paper.**

---

The following change log is included to help readers understand the modifications made throughout the author-reviewer discussion period.

**Nov. 21th** \
First revision based on the initial reviews from the reviewer. The modifications are highlighted in **blue**. The main differences are summarized as follows:
- **1. Added multiple LLM judges & human evaluation**\
In Section 5, we added evaluations from multiple LLM judges as well as human validation results.
- **2. Added experiments with the LLaMA model**\
In Section 4.1, we added results using the LLaMA-3.1-8B model. The GPT-5-nano baseline results were moved to a smaller table.
- **3. Added ablation test regarding K/J and rollout number**\
In Section 5, we added ablation studies on mental-state and utterance sampling settings, as well as rollout simulation counts.
- **4. Added Ethics Statement**\
In the Ethics Statement, we added ethical considerations for social intelligence in LLMs and details on the data collection used for human evaluation.
- **5. Added related work**\
In Related Work, we add the reference suggested by Reviewer WHSg.

---

### Note · Authors · 2026-01-06

I have read and agree with the venue's withdrawal policy on behalf of myself and my co-authors.